# Sample Complexity of Interventional Causal Representation Learning

**Emre Acartürk**
Rensselaer Polytechnic Institute

**Burak Varıcı**[*]
Carnegie Mellon University

**Karthikeyan Shanmugam**
Google DeepMind

**Ali Tajer**
Rensselaer Polytechnic Institute

## Abstract

Consider a data-generation process that transforms low-dimensional *latent* causally-related variables to high-dimensional *observed* variables. Causal representation learning (CRL) is the process of using the observed data to recover the latent causal variables and the causal structure among them. Despite the multitude of identifiability results under various interventional CRL settings, the existing guarantees apply exclusively to the *infinite-sample* regime (i.e., infinite observed samples). This paper establishes the first sample-complexity analysis for the finite-sample regime, in which the interactions between the number of observed samples and probabilistic guarantees on recovering the latent variables and structure are established. This paper focuses on *general* latent causal models, stochastic *soft* interventions, and a linear transformation from the latent to the observation space. The identifiability results ensure graph recovery up to ancestors and latent variables recovery up to mixing with parent variables. Specifically, $\mathcal{O}((\log \frac{1}{\delta})^4)$ samples suffice for latent graph recovery up to ancestors with probability $1 - \delta$, and $\mathcal{O}((\frac{1}{\epsilon} \log \frac{1}{\delta})^4)$ samples suffice for latent causal variables recovery that is $\epsilon$ close to the identifiability class with probability $1 - \delta$.

## 1 Introduction

The observed data generated in a wide range of technological, social, and biological domains has high-dimensional, complex, and often unexplainable structures. Nevertheless, sometimes, such complex structures can be explained by latent generating factors that often form lower-dimensional structures. The field of causal representation learning (CRL) is motivated by this premise and focuses on disentangling the *causally*-related latent generating factors that underlie a given high-dimensional observable dataset. In particular, given a high-dimensional dataset, the objective of CRL is to learn (i) the underlying latent causal variables and (ii) the causal relationships among the latent variables. The learned representations then provide an explainable structure for the observed data and facilitate informed reasoning for the downstream tasks [1].

Formally, CRL consists of a data generation and data transformation pipeline as follows. There exists a set of high-level latent variables $Z \in \mathbb{R}^n$ that are causally related. The causal interactions are captured by a directed acyclic graph (DAG) $\mathcal{G}$. The latent variables $Z$ go through an unknown transformation $g$ and generate the observed variables $X \in \mathbb{R}^d$, i.e., $X = g(Z)$. The objective of CRL is to use the observed variables $X$ and learn the latent graph $\mathcal{G}$ and latent variables $Z$. The primary question facing CRL is *identifiability*, which refers to establishing the conditions under which recovering $Z$ and $\mathcal{G}$ are possible.

---

[*]Work was done while BV was a Ph.D. student at Rensselaer Polytechnic Institute.

38th Conference on Neural Information Processing Systems (NeurIPS 2024).

Identifiability is known to be impossible without proper inductive biases or additional supervision [2, 3]. One approach to address this issue is performing *interventions* – leading to *interventional* CRL – which has gained significant recent attention [4, 5, 6, 7, 8, 9, 10]. Specifically, interventions on latent causal variables create additional statistical diversity in the observed data. Accessing to enough interventional environments enables identifiability [1]. Despite the recent advances in establishing identifiability guarantees for interventional CRL, all the existing guarantees focus on the asymptotic *infinite-sample* regime, where one assumes access to an infinite number of observable samples $X$.

In this paper, we establish conditions for **finite-sample non-asymptotic** identifiability and performance guarantees for interventional CRL. We focus on CRL under the *general* (non-parametric) latent causal models, *soft* interventions, and linear transformations from the latent to the observation space. This setting is well-studied in the infinite-sample regime [4, 5, 7, 11], where the existing identifiability results show that the causal graph can be recovered up to ancestral nodes, and latent variables can be learned up to mixing with their ancestors. In this paper, we provide the probabilistic finite-sample counterparts of these identifiability guarantees and establish the first sample complexity analysis for interventional CRL in the finite-sample regime.

Our CRL approach falls in the category of score-based CRL [5], based on which our sample complexity analysis consists of two steps. In the first step, we delineate a general sample complexity for any desired consistent score estimator. Subsequently, we specialize these general results by adopting the reproducing kernel Hilbert space (RKHS)-based score estimator [12]. We establish the following identifiability guarantees for any desired pair of constants $\epsilon, \delta \in \mathbb{R}_+$.

- **Latent graph recovery:** Using RHKS-based score estimator, $\mathcal{O}((\log \frac{1}{\delta})^4)$ samples suffice to recover the transitive closure of the latent graph with probability at least $1 - \delta$.

- **Latent variables recovery:** Using the same score estimator, $\mathcal{O}((\frac{1}{\epsilon} \log \frac{1}{\delta})^4)$ samples suffice to ensure that the mean squared error in the estimated latent causal variables is at most $\epsilon^2$ with probability at least $1 - \delta$.

- **Dependence on model dimension:** We further establish the explicit and implicit dependence of sample complexity on the dimensions of the latent and observable spaces, $n$ and $d$, respectively. The precise characterizations of these expressions involve additional model-dependent constants specified in Section 5.

- **Improved guarantees:** Finally, we note that our latent variables recovery result, where we show recovery up to mixing with *parents*, which improves upon the results in existing literature that guarantee recovering up to mixing with *ancestors*.

**Methodology.** We offer novel finite-sample CRL algorithms and characterize the sample complexity guarantees achievable by these algorithms. We design our algorithms using the properties of *score functions* (i.e., the gradient of the log density). This approach is inspired by the score-based CRL framework [5, 8]. However, the algorithms are significantly different, led by the need to integrate finite-sample score estimation routines and their imperfections. In our analysis, first, we provide sample complexity upper bounds for a score-based framework that uses any desired *consistent* score difference estimator. Then, we adopt a specific score estimator [12] and provide explicit sample complexity guarantees.

**Related work.** All the existing studies on interventional CRL focus on the infinite-sample regime. We briefly review the studies that adopt a similar CRL model, i.e., *linear* transformations and *single-node soft* interventions. In this setting, complete identifiability results for linear latent models are established in [7]. Similar results are shown for nonlinear latent causal models in [4, 11] and for linear non-Gaussian models in [13]. In the most closely related setting to ours, identifiability up to ancestors is shown to be possible using soft interventions on general latent causal models [8]. Other studies on interventional CRL include using *do* interventions for polynomial transformations [6, 14] and hard interventions for general transformations [10, 8]. On a partially related problem, error rates for using score functions on *observed* causal discovery are provided in [15].

**Notations.** For $n \in \mathbb{N}$, we define $[n] \triangleq \{1, \ldots, n\}$. Vectors are represented by lower case bold letters, and element $i$ of vector $\mathbf{v}$ is denoted by $\mathbf{v}_i$. Matrices are represented by upper case bold letters, and we denote row $i$ and column $j$ of matrix $\mathbf{A}$ by $\mathbf{A}_{i,:}$ and $\mathbf{A}_{:,j}$, respectively. The row permutation matrix for permutation $\pi$ of $[n]$ is denoted by $\mathbf{P}_\pi$. Random variables and their realizations are presented by upper and lower case letters, respectively. We denote the Moore-Penrose pseudoinverse

of a matrix $\mathbf{A}$ by $\mathbf{A}^\dagger$. Given a symmetric matrix $\mathbf{A} \in \mathbb{R}^{d \times d}$, we denote the vector of eigenvalues of $\mathbf{A}$ ordered in ascending order by $\boldsymbol{\lambda}(\mathbf{A}) \in \mathbb{R}^d$ and the matrix of eigenvectors by $\mathbf{Q}(\mathbf{A}) \in \mathbb{R}^{d \times d}$ such that $\mathbf{A} = \mathbf{Q}(\mathbf{A}) \cdot \mathrm{diag}(\boldsymbol{\lambda}(\mathbf{A})) \cdot \mathbf{Q}(\mathbf{A})^\top$. For any matrix $\mathbf{A}$, we denote the rank, column, and null spaces of $\mathbf{A}$ by $\mathrm{rank}(\mathbf{A})$, $\mathrm{col}(\mathbf{A})$, and $\mathrm{null}(\mathbf{A})$, respectively.

## 2 Finite-sample data-generation process

The existing studies on analyzing the identifiability guarantees of CRL inevitably necessitate the availability of an infinite number of samples as otherwise, perfect identifiability is rendered impossible. In this paper, our objective is identifiability analysis assuming only **a finite number of samples** are available – a significant departure from the infinite-sample regime. In this section, we specify a general data-generating process consisting of a latent causal space and an unknown linear transformation that maps the latent variables to observed variables.

**Latent causal model.** We have a latent space of $n$ causally related random variables. We denote the latent causal variables by $Z \triangleq [Z_1, \ldots, Z_n]^\top$. The causal relationships among the variables of $Z$ are represented by a directed acyclic graph $\mathcal{G}$ with $n$ nodes where the $i$-th node represents $Z_i$. We denote the parents, children, and ancestors of a node $i \in [n]$ in $\mathcal{G}$ by $\mathrm{pa}(i)$, $\mathrm{ch}(i)$, and $\mathrm{an}(i)$, respectively. We denote the probability density function (pdf) of $Z$ by $p_Z$, which is assumed to be well-defined without any zeros over lower-dimensional manifolds and to have full support on $\mathbb{R}^n$. Given the DAG $\mathcal{G}$, $p_Z$ factorizes according to

$$p_Z(z) = \prod_{i \in [n]} p_i(z_i \mid z_{\mathrm{pa}(i)}) \,. \tag{1}$$

We assume that the conditional pdfs $\{p_i : i \in [n]\}$ are continuously differentiable with respect to $z$. We call a permutation $\pi = (\pi_1, \ldots, \pi_n)$ of $[n]$ a *valid causal order* if for all $i, j \in [n]$, $i \in \mathrm{pa}(j)$ implies $\pi_i < \pi_j$. Without loss of generality, we assume that $(1, \ldots, n)$ is a valid causal order.

**Transformation model.** The latent variables $Z$ are mapped to the observed variables $X \triangleq [X_1, \ldots, X_d]^\top$ through an unknown linear transformation $\mathbf{G} \in \mathbb{R}^{d \times n}$, where $d \geq n$. Specifically,

$$X = \mathbf{G} \cdot Z \,, \tag{2}$$

where $\mathbf{G}$ has full column rank. We denote the pdf of $X$ by $p_X$. Note that owing to the generation process of $X$, the pdf $p_X$ is supported on an $n$-dimensional subspace embedded in $\mathbb{R}^d$. We denote the $L^2$ norm of any function $f$ with finite variance under $p_X$ by $\|f\|_{p_X}^2 \triangleq \mathbb{E}_{p_X} \|f(x)\|_2^2$.

**Intervention model.** We consider CRL under interventions, and in particular focus on *soft* interventions, as the most general form of interventions. Hence, in addition to the observational model specified by (2), we consider $n$ single-node interventional environments $\{\mathcal{E}^m : m \in [n]\}$. We assume that the node intervened in the environment $\mathcal{E}^m$ is *unknown* and denote it by $I^m$. Leaving one node unintervened renders identifiability impossible [7]. Hence, we inevitably have $\{I^m : m \in [n]\} = [n]$.

Applying a soft intervention on node $i$ changes its causal mechanism specified by the observational conditional pdf $p_i(z_i \mid z_{\mathrm{pa}(i)})$ to a distinct interventional conditional pdf $q_i(z_i \mid z_{\mathrm{pa}(i)})$. The conditional pdfs $\{q_i : i \in [n]\}$ are assumed to be continuously differentiable with respect to $z$. Subsequently, the pdf of $Z$ in the interventional environment $\mathcal{E}^m$, denoted by $p^m$, factorizes according to

$$p_Z^m(z) = q_\ell(z_\ell \mid z_{\mathrm{pa}(\ell)}) \cdot \prod_{i \neq \ell} p_i(z_i \mid z_{\mathrm{pa}(i)}) \,, \quad \text{where} \quad \ell = I^m \,. \tag{3}$$

To distinguish the observational and interventional data, we denote the latent and observed random variables in environment $\mathcal{E}^m$ by $Z^m$ and $X^m$, respectively. Interventions do not alter the transformation matrix $\mathbf{G}$, indicating that similarly to (2), we have $X^m = \mathbf{G} \cdot Z^m$. Throughout the paper, we adopt the convention that $\mathcal{E}^0$ refers to the observational environment.

**Score functions.** The *score function* associated with a pdf is defined as the gradient of its logarithm. We denote the score functions associated with the observational distributions of $Z$ and $X$ by $s_Z$ and $s_X$, respectively, i.e.,

$$s_Z(z) \triangleq \nabla_z \log p_Z(z) \,, \quad \text{and} \quad s_X(x) \triangleq \nabla_x \log p_X(x) \,. \tag{4}$$

Similarly, we denote the score functions associated with the distributions of $Z$ and $X$ in environment $\mathcal{E}^m$ for $m \in [n]$ by $\boldsymbol{s}_Z^m$ and $\boldsymbol{s}_X^m$, respectively, i.e.,

$$\boldsymbol{s}_Z^m(z) \triangleq \nabla_z \log p_Z^m(z) , \quad \text{and} \quad \boldsymbol{s}_X^m(x) \triangleq \nabla_x \log p_X^m(x) . \tag{5}$$

In our algorithm and analysis, we will analyze the score variations across different interventions. For this purpose, we define the score difference functions of $Z$ and $X$ between environments $\mathcal{E}^m$ and $\mathcal{E}^0$ as

$$\boldsymbol{d}_Z^m(z) \triangleq \boldsymbol{s}_Z^m(z) - \boldsymbol{s}_Z(z) , \quad \text{and} \quad \boldsymbol{d}_X^m(x) \triangleq \boldsymbol{s}_X^m(x) - \boldsymbol{s}_X(x) , \quad \forall m \in [n] . \tag{6}$$

To ensure that interventions affect the target node and its parents distinctly, we adopt the following assumption. This assumption holds for a large class of models, e.g., additive noise models under stochastic hard interventions [5, Lemma 2].

**Assumption 1** ([5, Assumption 1]). *For any $m \in [n]$, and for all $k \in \mathrm{pa}(I^m)$, the term $\left([\boldsymbol{d}_Z^m]_k / [\boldsymbol{d}_Z^m]_{I^m}\right)$ is not a constant function of $z$.*

**Finite sample data.** We assume that we have only $N$ data samples under each of the environments. We denote the $N$ samples of $Z^m$ and $X^m$ by

$$\{z_j^m : j \in [N]\} , \quad \text{and} \quad \{x_j^m : j \in [N]\} , \quad \forall m \in \{0\} \cup [n] . \tag{7}$$

Accordingly, for each sample $j$, we concatenate all latent and observational samples under different environments to create sample matrices

$$\boldsymbol{Z}_j \triangleq [z_j^0, \dots, z_j^n] , \quad \text{and} \quad \boldsymbol{X}_j \triangleq [x_j^0, \dots, x_j^n] , \quad \forall j \in [N] . \tag{8}$$

Finally, we define the set of samples $\mathcal{X}_N \triangleq \{\boldsymbol{X}_j : j \in [N]\}$ and $\mathcal{Z}_N \triangleq \{\boldsymbol{Z}_j : j \in [N]\}$.

## 3 Finite-sample identifiability objectives

The CRL framework specified in Section 2 has two objectives: use the $N$ observational samples $\mathcal{X}_N = \{\boldsymbol{X}_j : j \in [N]\}$ to recover the causal graph $\mathcal{G}$ and the latent causal variables $\mathcal{Z}_N = \{\boldsymbol{Z}_j : j \in [N]\}$. In this section, we formalize the inference rules and their associated fidelity metrics for recovering $\mathcal{G}$ and $\mathcal{Z}_N$.

**Latent graph recovery.** Define $\mathscr{G}$ as the set of all DAGs on $n$ nodes. We define $\hat{\mathcal{G}}$ as a generic estimator of the latent graph $\mathcal{G}$, i.e.,

$$\hat{\mathcal{G}} \colon \left(\mathbb{R}^{d \times (n+1)}\right)^N \to \mathscr{G} . \tag{9}$$

To quantify the accuracy of the estimate $\hat{\mathcal{G}}(\mathcal{X}_N)$, we provide the following probably approximately correct (PAC) guarantee, which is the probabilistic counterpart of the standard *identifiability up to ancestor* definition in the interventional CRL literature [5, 7, 11].

**Definition 1** ($\delta$-PAC graph recovery). *Graph estimate $\hat{\mathcal{G}}(\mathcal{X}_N)$ achieves $\delta$–PAC latent graph recovery if, with probability at least $1 - \delta$, the transitive closures of $\hat{\mathcal{G}}(\mathcal{X}_N)$ and $\mathcal{G}$ are isomorphic.*

**Latent variables recovery.** We investigate a two-step estimator for the latent variables $\mathcal{Z}_N$. First, we define a generic estimator for the pseudoinverse of $\mathbf{G}$ as

$$\mathbf{H} \colon \left(\mathbb{R}^{d \times (n+1)}\right)^N \to \mathbb{R}^{n \times d} . \tag{10}$$

Then, given an estimate $\mathbf{H}(\mathcal{X}_N)$ for $\mathbf{G}^\dagger$, we estimate $\{\boldsymbol{Z}_j : j \in [N]\}$ according to

$$\hat{\boldsymbol{Z}}_j = \mathbf{H}(\mathcal{X}_N) \cdot \boldsymbol{X}_j , \qquad \forall j \in [N] . \tag{11}$$

We provide the following definition to quantify the fidelity of these estimates with respect to the ground truth variables.

**Definition 2** ($(\epsilon, \delta)$–PAC variables recovery). *The estimate $\mathbf{H}(\mathcal{X}_N)$ achieves $(\epsilon, \delta)$–PAC latent variables recovery if the estimated causal variables $\{\hat{\boldsymbol{Z}}_j : j \in [N]\}$ satisfy*

$$\hat{\boldsymbol{Z}}_j = \mathbf{P}_I \cdot (\mathbf{C}_{\mathrm{pa}} + \mathbf{C}_{\mathrm{err}}) \cdot \boldsymbol{Z}_j , \tag{12}$$

*where for all $i \notin \overline{\mathrm{pa}}(j)$, $\mathbf{C}_{\mathrm{pa}}$ satisfies $(\mathbf{C}_{\mathrm{pa}})_{i,j} = 0$, and, with probability at least $1 - \delta$, we have $\|\mathbf{C}_{\mathrm{err}}\|_2 \leq \epsilon$.*

We note that Definition 2 is the probabilistic counterpart of the standard *identifiability up to ancestor* definition in interventional CRL literature.

**Noisy score model.** The score-based CRL framework uses properties of score function differences to build the estimators for the latent graph and variables. Therefore, when we only have access to finite data, we need to estimate the score functions. We denote a generic score function estimator for environment $\mathcal{E}^m$ by

$$\hat{s}_X^m(x; \mathcal{X}_N) \colon \mathbb{R}^d \times \left(\mathbb{R}^{d \times (n+1)}\right)^N \to \mathbb{R}^d \, . \tag{13}$$

When the dependence is clear from the context, we will drop the explicit dependence of $\hat{s}_X^m$ on $\mathcal{X}_N$. Similarly to (6), we define the estimated score difference functions

$$\hat{d}_X^m(x; \mathcal{X}_N) \triangleq \hat{s}_X^m(x; \mathcal{X}_N) - \hat{s}_X(x; \mathcal{X}_N) \, , \quad \forall m \in [n] \, . \tag{14}$$

In this paper, we focus on *consistent* score difference estimators $\{\hat{d}_X^m \, : \, m \in [n]\}$, i.e., they satisfy convergence in probability. Specifically, for any $\epsilon, \delta > 0$, there exists $N(\epsilon, \delta) \in \mathbb{N}$ such that

$$\mathbb{P}\left( \max_{m \in [n]} \left\| \hat{d}_X^m(\cdot; \mathcal{X}_N) - d_X^m \right\|_{p_X} \le \epsilon \right) \ge 1 - \delta \, , \qquad \forall N \ge N(\epsilon, \delta) \, . \tag{15}$$

Notably, we note that many score estimators are known to be consistent [12, 16, 17, 18].

## 4 Finite-sample CRL algorithms

In this section, we design finite-sample interventional CRL algorithms through which we establish finite-sample identifiability guarantees as well as bounds on the associated sample complexities. Our algorithms fall within the score-based category of CRL algorithms [5]. The main intuition of this framework is that score functions in the observed space contain all the information needed for recovering the latent graph $\mathcal{G}$ and the inverse transform $\mathbf{G}^\dagger$. Specifically, two metrics are pivotal for retrieving the latent space:

1. **Score differences:** As shown in Lemma 1, the nonzero entries of the latent score differences $d_Z^m$ encode the graph structure and the observed score differences $d_X^m$ are generated using the inverse transform $\mathbf{G}^\dagger$.

2. **Score difference correlations:** As shown in Lemma 2, the column space of the correlation matrix of the score differences contains crucial information about the latent graph and the inverse transform, which we will leverage to form estimates for the latent graph $\mathcal{G}$ and the inverse transform $\mathbf{G}^\dagger$.

To proceed, for $m \in [n]$ we denote the correlation matrices of $d_X^m$ and $\hat{d}_X^m$ by

$$\mathbf{R}_X^m \triangleq \mathbb{E}_{p_X}\left[ d_X^m(x) \cdot (d_X^m(x))^\top \right] \, , \quad \text{and} \quad \hat{\mathbf{R}}_X^m \triangleq \mathbb{E}_{p_X}\left[ \hat{d}_X^m(x) \cdot (\hat{d}_X^m(x))^\top \right] \, . \tag{16}$$

Next, we formalize two critical properties of score differences and their correlation matrix. Specifically, the following lemma states that the sparsity pattern of the latent score differences $d_Z^m$ exposes the structure of the latent graph, and the observed score differences $d_X^m$ preserve this information through the pseudoinverse of $\mathbf{G}$.

**Lemma 1** ([5]). *Score function differences in the latent space satisfy, for all $m \in [n]$,*

$$\forall z \in \mathbb{R}^n : \qquad [d_Z^m(z)]_i = 0 \iff i \notin \overline{\mathrm{pa}}(I^m) \, . \tag{17}$$

*Furthermore, the score differences in the observed domain are given by*

$$d_X^m(x) = (\mathbf{G}^\dagger)^\top \cdot d_Z^m(z) \, , \quad x = \mathbf{G} \cdot z \, . \tag{18}$$

The next lemma specifies that the structure of the column space of correlation matrix $\mathbf{R}_X^m$ is heavily constrained by the graph structure and the inverse transformation $\mathbf{G}^\dagger$.

**Lemma 2.** *For any $m \in [n]$, we have $\mathsf{col}(\mathbf{R}_X^m) \subseteq \mathsf{span}\left\{ (\mathbf{G}^\dagger)^\top_{:,i} \, : \, i \in \overline{\mathrm{pa}}(I^m) \right\}$.*

While these two properties enable recovering the latent graph and variables, the ground truth score functions $d_Z^m(z)$ and correlation matrix $\mathbf{R}_X^m$ are unknown. We only have access to their noisy counterparts, as estimated from the observed data. In our algorithms, we use methods with soft decision rules that can counter the effects of the errors introduced by the score estimation procedure. These are facilitated by the following three key definitions.

---

**Algorithm 1** Causal order estimation

---

1: **Input:** $\hat{\mathbf{R}}^m$ for all $m \in [n]$, $\eta \geq 0$
2: $\mathcal{V}_n \leftarrow \{1, \ldots, n\}$                                          ▷ remaining unordered set
3: **for** $t \in (n, \ldots, 2)$ **do**
4:     **for** $k \in \mathcal{V}_t$ **do**
5:         **if** $\dim \mathrm{col}(\sum_{m \in \mathcal{V}_t \setminus \{k\}} \hat{\mathbf{R}}^m; \eta) = t - 1$ **then**
6:             $\pi_t \leftarrow k$                                          ▷ $I^k$ has no ancestors in $I^{\mathcal{V}_t}$
7:             $\mathcal{V}_{t-1} \leftarrow \mathcal{V}_t \setminus \{k\}$           ▷ remove the identified node from unordered set
8:             **break** $k$ loop
9: $\pi_1 \leftarrow m$ for $m \in \mathcal{V}_1$
10: **Return** $\pi$

---

---

**Algorithm 2** Graph estimation

---

1: **Input:** $\hat{\mathbf{R}}^m$ for all $m \in [n]$, $\pi$, $\eta \geq 0$, $\gamma \geq 0$
2: Initialize $\hat{\mathcal{G}}$ with empty graph over nodes $\{\pi_1, \ldots, \pi_n\}$
3: Construct the $\mathcal{V}_t$ sets of Algorithm 1 for all $t \in [n]$ using $\pi$
4: **for** $t \in (n-1, \ldots, 1)$ **do**
5:     **for** $j \in (t+1, n)$ **do**
6:         $\mathcal{M}_{t,j} \leftarrow \mathcal{V}_j \setminus (\{\pi_t\} \cup \hat{\mathrm{ch}}(\pi_t))$                    ▷ set for determining whether $\pi_t \to \pi_j$
7:         **if** $\mathrm{col}(\sum_{m \in \mathcal{V}_t} \hat{\mathbf{R}}^m; \eta) \perp_\gamma \mathrm{null}(\sum_{m \in \mathcal{M}_{t,j}} \hat{\mathbf{R}}^m; \eta)$ **then**
8:             Add $\pi_t \to \pi_j$ and $\pi_t \to l$ to $\hat{\mathcal{G}}$ for all $l \in \hat{\mathrm{ch}}(\pi_j)$ in $\hat{\mathcal{G}}$
9: **Return** $\hat{\mathcal{G}}$

---

**Definition 3** (Approximate column space). *We define the $\eta$-approximate column space of a positive semidefinite matrix $\mathbf{A} \in \mathbb{R}^{d \times d}$, denoted by $\mathrm{col}(\mathbf{A}; \eta)$, as the span of the eigenvectors of $\mathbf{A}$ associated with the eigenvalues that are strictly greater than $\eta$.*

**Definition 4** (Approximate null space). *We define the $\eta$-approximate null space of $\mathbf{A}$, denoted by $\mathrm{null}(\mathbf{A}; \eta)$, as the orthogonal complement of $\mathrm{col}(\mathbf{A}; \eta)$.*

**Definition 5** (Approximate subspace orthogonality). *We say two subspaces $\mathcal{A}$ and $\mathcal{B}$ with orthonormal bases $\mathbf{A}$ and $\mathbf{B}$ are $\gamma$-approximately orthogonal, denoted by $\mathcal{A} \perp_\gamma \mathcal{B}$, if $\|\mathbf{B}^\top \mathbf{A}\|_2 \leq \gamma$.*

In these definitions, setting $\eta = \gamma = 0$ yields the standard definitions of column and null spaces and orthogonality. Next, we describe our algorithms for latent graph recovery (Algorithms 1 and 2) and latent variables recovery (Algorithm 3). We note that the algorithms for latent graph and latent variable recovery algorithms are fully decoupled, enabling the independent recovery of the graph and the latent variables.

**Algorithm 1 – Causal order estimation.** In this algorithm, we estimate a permutation $\pi$ of $[n]$ such that $I \circ \pi$ is a valid causal order. This serves as an intermediate step for estimating the latent graph. For this purpose, we use the noisy correlation matrices $\{\hat{\mathbf{R}}_X^m : m \in [n]\}$ to identify the *leaf* nodes with no children. In particular, the key property is that when $I^k$ is a leaf node, by carefully selecting the threshold $\eta$, the approximate column space of the term $(\sum_{m \in [n]} \hat{\mathbf{R}}_X^m - \hat{\mathbf{R}}_X^k)$ has dimension $n - 1$. Precisely, with high probability, we have

$$\dim \mathrm{col}\Big( \sum_{m \in [n] \setminus \{k\}} \hat{\mathbf{R}}_X^m \; ; \; \eta \Big) = n - 1 \quad \Longleftrightarrow \quad I^k \text{ is a leaf node} . \tag{19}$$

After finding a leaf node, we iteratively identify the *youngest* node among the remaining set of nodes. Leveraging this, we construct the permutation $\pi$, which consists of nodes ordered from the eldest to the youngest. In Lemma 3, we show that $I \circ \pi$ is a valid causal order with high probability.

**Algorithm 2 – Latent graph estimation.** In Algorithm 2, we use the causal order found in Algorithm 1 and correlation matrices $\{\hat{\mathbf{R}}_X^m : m \in [n]\}$ to form a graph estimate $\hat{\mathcal{G}}$. We build this graph iteratively by considering a candidate edge $\pi_t \to \pi_j$ for all possible $(t, j)$ pairs starting from a leaf node $t$. The key property that we leverage is that we can form two subsets $\mathcal{V}_t$ and $\mathcal{M}_{t,j}$ of $[n]$ using $\pi$,

---
**Algorithm 3** Inverse transform estimation
---

1: **Input:** $\hat{\mathbf{R}}^m$ for all $m \in [n]$, $\eta \geq 0$
2: $\mathbf{H} \leftarrow \mathbf{0}_{n \times d}$
3: **for** $t \in (1, \ldots, n)$ **do**
4:      $\mathcal{C} \leftarrow \mathsf{col}(\hat{\mathbf{R}}^{\pi_t}; \eta)$
5:      $\mathbf{H}_{\pi_t,:} \leftarrow \mathbf{v}$ for $\mathbf{v} \in \mathcal{C}$ and $\|\mathbf{v}\|_2 = 1$
6: **Return H**

---

$t$, and $j$ such that, for sufficiently small $\gamma$, with high probability the following approximation holds.

$$\mathsf{col}\Big( \sum_{m \in \mathcal{V}_t} \hat{\mathbf{R}}_X^m \; ; \; \eta \Big) \perp_\gamma \mathsf{null}\Big( \sum_{m \in \mathcal{M}_{t,j}} \hat{\mathbf{R}}_X^m \; ; \; \eta \Big) \iff I^{\pi_t} \to I^{\pi_j} \text{ in } \mathcal{G} \; . \tag{20}$$

We check each edge $\pi_t \to \pi_j$ and add the detected edges to the estimated graph $\hat{\mathcal{G}}$. In Theorem 1, we show that this procedure guarantees a PAC latent graph recovery.

**Algorithm 3 – Inverse transform estimation.** In Algorithm 3, we build our inverse transform estimate $\mathbf{H}$ one row at a time. The key property we use is that for any $m \in [n]$ and unit vector $\mathbf{v} \in \mathsf{col}(\hat{\mathbf{R}}_X^m)$, the following error term is small with high probability

$$\Big\| \mathbf{v}^\top \cdot X - \sum_{i \in \overline{\mathrm{pa}}(I^m)} \big( \mathbf{v}^\top \cdot \mathbf{G} \big)_i \cdot Z_i \Big\|_2 \; . \tag{21}$$

In other words $\mathbf{v}^\top \cdot X$ is approximately equal to $Z_{I^m}$ up to mixing with $\{Z_i : i \in \mathrm{pa}(I^m)\}$, which conforms to our latent variables recovery objective. Note that in the noise-free setting, an exact equality holds due to Lemma 2 and property $\mathbf{G}^\dagger \cdot \mathbf{G} = \mathbf{I}_n$. Based on this observation, in our algorithm, we construct our inverse transform estimate by setting row $m$ of $\mathbf{H}$ to a unit vector $\mathbf{v} \in \mathsf{col}(\hat{\mathbf{R}}_X^m)$ for all $m \in [n]$. In Theorem 2, we show that Algorithm 3 achieves a PAC latent variables recovery.

## 5 Sample complexity analysis

In this section, we analyze the sample complexity of our CRL algorithm to establish an achievable sample complexity for CRL.

**Threshold selection.** A critical step in our algorithm designs is choosing thresholds $\eta$ and $\gamma$ specified in Definitions 3–5. These thresholds determine the approximate ranks of noisy matrices and approximate orthogonality between subspaces, respectively. Specifically, we select $\eta$ such that the approximate rank of $\hat{\mathbf{R}}_X^m$ tracks that of $\mathbf{R}_X^m$ with high probability. To proceed, we denote the minimum nonzero eigenvalue among arbitrary $\mathbf{R}_X^m$ sums by

$$\eta^* \triangleq \min_{\mathcal{M} \subseteq [n]} \min \Big\{ \boldsymbol{\lambda}_i \Big( \sum_{m \in \mathcal{M}} \mathbf{R}_X^m \Big) : i \in [d] \, , \, \boldsymbol{\lambda}_i \Big( \sum_{m \in \mathcal{M}} \mathbf{R}_X^m \Big) \neq 0 \Big\} \, , \tag{22}$$

where $\boldsymbol{\lambda}(\mathbf{A}) \in \mathbb{R}^d$ denotes the vector of eigenvalues of $\mathbf{A}$. In our algorithms, we let $\eta \in (0, \eta^*)$ and show that this choice ensures that eigenvalues of the null and column spaces can be separated via $\eta$. For the approximate orthogonality test in Definition 5, we show that $\gamma^*$ defined below serves as a bound on how close the non-orthogonal column spaces of $\mathbf{R}_X^m$ sums become orthogonal.

$$\gamma^* \triangleq \min_{i \in [n]} \big\| \mathbf{G}^\dagger{}_{i,:} \big\|_2 \cdot \big\| \mathbf{G}_{:,i} \big\|_2 \, . \tag{23}$$

Similarly, we let $\gamma \in (0, \gamma^*)$ in our algorithms. For any choice of $\eta$ and $\gamma$, we define $\eta_* \triangleq \min\{\eta, \eta^* - \eta\}$ and $\gamma_* \triangleq \min\{\gamma, \gamma^* - \gamma\}$. We note that we do not have to know $\eta^*$ and $\gamma^*$ a priori, and can include routines to estimate them in practice. These estimates do not have to be highly accurate since even rough estimates suffice to choose reliable thresholds $\eta$ and $\gamma$. Specifically, based on estimates for $\eta^*$ and $\gamma^*$, we can choose "safe" thresholds for $\eta$ and $\gamma$, e.g., one-fourth of the estimates for $\eta^*$ and $\gamma^*$, so that (i) with high probability we satisfy the requirement $\eta \in (0, \eta^*)$ and $\gamma \in (0, \gamma^*)$, and (ii) we can avoid collecting excessive samples and compromising the sample complexity bounds. We provide the details of constructing such estimates of $\eta^*$ and $\gamma^*$ in Appendix G.

**Sample complexity results.** We provide two sets of sample complexity analyses for the latent graph and the latent variables. We estimate the latent graph in two steps: (i) causal order estimation

(Algorithm 1) and (ii) latent graph estimation (Algorithm 2). Algorithm 1 only employs approximate rank tests. Therefore, we first show that the approximate rank of the sum of correlation matrices $\{\hat{\mathbf{R}}_X^m : m \in [n]\}$ tracks that of $\{\mathbf{R}_X^m : m \in [n]\}$ with high probability. To proceed, we define the following constants that appear repeatedly in our analysis:

$$\beta \triangleq \Big( 4 \max_{m \in [n]} \big\| \boldsymbol{d}_X^m \big\|_{p_X} \Big)^{-1} \qquad \text{and} \qquad \beta_{\min} \triangleq 2 \min_{m \in [n]} \big\| \boldsymbol{d}_X^m \big\|_{p_X} . \tag{24}$$

**Lemma 3.** *Let $\eta \in (0, \eta^*)$. For any $\delta > 0$, $N_{\mathrm{rank}}(\delta)$ samples suffice to ensure that with probability at least $1 - \delta$,*

$$\forall \mathcal{M} \subseteq [n] : \qquad \mathsf{dim\,col}\Big( \sum_{m \in \mathcal{M}} \hat{\mathbf{R}}_X^m ; \eta \Big) = \mathsf{rank}\Big( \sum_{m \in \mathcal{M}} \mathbf{R}_X^m \Big) , \tag{25}$$

*where*

$$N_{\mathrm{rank}}(\delta) \triangleq N\Big( \min\Big\{ \frac{\beta\eta_*}{n}, \beta_{\min} \Big\}, \delta \Big) . \tag{26}$$

Note that the function $N(\cdot, \cdot)$ is an inherent property of the score difference estimator specified in (15), and it is monotonically decreasing in its second argument. Using this lemma, we can show that Algorithm 1 returns a permutation $\pi$ such that $I \circ \pi$ is a valid causal order with high probability.

**Lemma 4.** *Let $\eta \in (0, \eta^*)$. Under Assumption 1, for any $\delta > 0$, $N_{\mathrm{rank}}(\delta)$ samples suffice to ensure that with probability at least $1 - \delta$, $I \circ \pi$ is a valid causal order, where $\pi$ is the output of Algorithm 1.*

Next, we show that Algorithm 2 achieves latent graph recovery with high probability.

**Theorem 1** (Sample complexity – Graph). *Let $\eta \in (0, \eta^*)$ and $\gamma \in (0, \gamma^*)$. Under Assumption 1, for any $\delta > 0$, $N_{\mathcal{G}}(\delta)$ samples suffice to ensure that the output $\hat{\mathcal{G}}(\mathcal{X}_N)$ collectively generated by Algorithms 1 and 2 satisfies $\delta$–PAC graph recovery, where*

$$N_{\mathcal{G}}(\delta) \triangleq N\left( \min\Big\{ \frac{\beta\eta_*}{n}, \frac{\beta\eta_*\gamma_*}{2n}, \beta_{\min} \Big\}, \delta \right) . \tag{27}$$

We note that constants $\beta$ and $\beta_{\min}$ are sample-independent. Then, (27) specifies the sample complexity as a function of the hyperparameters $\eta, \gamma$, target reliability $\delta \in (0, 1)$, as well as the latent dimension $n$. Next, we state the complementary result for recovering the latent variables.

**Theorem 2** (Sample complexity – Variables). *Let $\eta \in (0, \eta^*)$. For any $\epsilon > 0$ and $\delta > 0$, $N_Z(\epsilon, \delta)$ samples suffice to ensure that the output $\mathbf{H}(\mathcal{X}_N)$ of Algorithm 3 satisfies $(\epsilon, \delta)$–PAC causal variables recovery, where*

$$N_Z(\epsilon, \delta) \triangleq N\left( \min\Big\{ \frac{\epsilon\beta\eta_*}{\sqrt{n}\|\mathbf{G}\|_2}, \beta\eta_*, \beta_{\min} \Big\}, \delta \right) . \tag{28}$$

Theorems 1 and 2 collectively specify an extent of identifiability achievable in the finite-sample regime. Importantly, we note that since we can establish $(\epsilon, \delta)$–PAC identifiability guarantees for *any* vanishing $\epsilon, \delta > 0$ using finite samples, Algorithms 2 and 3 are *consistent* estimators of the latent graph and the inverse transformation up to the corresponding equivalence classes.

Finally, we note that the latent variables are recovered up to mixing with *parents*, as specified in Theorem 2. This result is a refinement of the existing latent variable recovery results under soft interventions in existing literature [5, 7], which recover the latent variables up to mixing with *ancestors*.

**RKHS-based score estimator.** So far, we have characterized the sample complexity for any consistent score difference estimator. Next, we specialize the results by adopting the RKHS-based score estimator of [12]. We adopt this particular choice since it has known non-asymptotic sample complexity properties. To our knowledge, it is the only score estimator equipped with such a guarantee. To use the sample complexity of this score estimator in our paper, we first show that it is consistent in the sense of (15). For the formal statement of the following property and its attendant assumptions, we refer to Appendix E.

**Lemma 5** (Informal). *Assume that $\boldsymbol{s}_X$ and $\boldsymbol{s}_X^m$ satisfy the conditions of the RKHS-based score estimator in [12]. Then, for any given $\delta$ and $\epsilon$, the convergence specified in (15) holds when*

$$N(\epsilon, \delta) \triangleq \Big( \max\Big\{ \frac{C}{\epsilon}, 2\sqrt{2}\kappa^2 \Big\} \Big)^4 \cdot \Big( \log\Big( \frac{8n}{\delta} \Big) \Big)^4 , \tag{29}$$

*where $\kappa$ and $C$ are sample independent constants that depends only on $p_X$, $p_X^m$ for $m \in [n]$ and the structure of the RKHS.*

We use Lemma 5 to customize the general sample complexity bounds in Theorems 1 and 2 to the RKHS-based score estimator. For this purpose, we first provide our sample complexity result for recovering the latent graph.

**Theorem 3** (RKHS-based sample complexity – Graph). *Let $\eta \in (0, \eta^*)$ and $\gamma \in (0, \gamma^*)$. Under Assumption 1 and the conditions of Lemma 5, $N_{\mathcal{G}}(\delta)$ samples suffice to ensure that $\hat{\mathcal{G}}(\mathcal{X}_N)$ collectively generated by Algorithms 1 and 2 satisfies $\delta$–PAC graph recovery, where*

$$N_{\mathcal{G}}(\delta) = \left( \max \left\{ \frac{C}{\epsilon_{\mathcal{G}}}, \ 2\sqrt{2}\kappa^2 \right\} \right)^4 \cdot \left( \log \frac{8n}{\delta} \right)^4, \quad \text{and} \quad \epsilon_{\mathcal{G}} \triangleq \min \left\{ \frac{\beta\eta_*}{n}, \frac{\beta\eta_*\gamma_*}{2n}, \beta_{\min} \right\}. \tag{30}$$

**Remark 1** (RKHS-based error bound – Graph). *Theorem 3 implies that using $N$ samples, the output $\hat{\mathcal{G}}(\mathcal{X}_N)$ of Algorithms 1 and 2 satisfies $\delta_{\mathcal{G}}(N)$–PAC graph recovery, where*

$$\delta_{\mathcal{G}}(N) \triangleq 8n \cdot \exp \left( -N^{1/4} \cdot \max \left\{ \frac{C}{\epsilon_{\mathcal{G}}}, \ 2\sqrt{2}\kappa^2 \right\} \right). \tag{31}$$

Similarly, we leverage Lemma 5 to specialize the general sample complexity in Theorem 2 to the RKHS-based score estimator.

**Theorem 4** (RKHS-based sample complexity – Variables). *Let $\eta \in (0, \eta^*)$. Under the conditions of Lemma 5, $N_Z(\epsilon, \delta)$ samples suffice to ensure that the output $\mathbf{H}(\mathcal{X}_N)$ of Algorithm 3 satisfies $(\epsilon, \delta)$–PAC causal variables recovery, where*

$$N_Z(\epsilon, \delta) = \left( \max \left\{ \frac{C}{\epsilon \cdot \epsilon_Z}, \ 2\sqrt{2}\kappa^2 \right\} \right)^4 \cdot \left( \log \left( \frac{8n}{\delta} \right) \right)^4, \tag{32}$$

*and*

$$\epsilon_Z \triangleq \min \left\{ \frac{\epsilon\beta\eta_*}{\sqrt{n}\|\mathbf{G}\|_2}, \ \beta\eta_*, \ \beta_{\min} \right\}. \tag{33}$$

We note that the results in Theorems 3 and 4 show a more explicit dependence of the sample complexity on the latent space dimension $n$. We observe that the sample complexity of latent graph recovery explicitly depends on $\delta$ and the latent dimension $n$ according to $\mathcal{O}((n \log \frac{n}{\delta})^4)$. Similarly, the sample complexity of latent variables recovery explicitly depends on $\epsilon$, $\delta$, and the latent dimension $n$ according to $\mathcal{O}((\frac{\sqrt{n}}{\epsilon} \log \frac{n}{\delta})^4)$. We note that constants $\beta$, $\beta_{\min}$, $\eta^*$, and $\gamma^*$ are model parameters, and their scaling behavior in terms of $n$ and $d$ are investigated numerically in the next section.

# 6 Experiments

In this section, we perform numerical assessments of our analyses to provide complementary insight into the sample complexity results of Section 5. Specifically, we evaluate the variations of the model constants with respect to problem dimensions $n$ and $d$. Furthermore, we also evaluate performance variations of the finite-sample algorithm in terms of the variations in the score estimation error. For this purpose, we focus on the mean squared error (MSE) of the score estimator, specified by $\mathbb{E}\|\hat{d}_X^m - d_X^m\|_{p_X}^2$. These evaluations facilitate a better understanding of the properties of the CRL problem that are not explicit in our theoretical queries.[2]

**Experimental details.** We consider problem dimensions $n \in \{3, 5, 10\}$ and $d \in \{n, 15\}$ and generate $\mathcal{G}$ using Erdős-Rényi model with density $0.5$ on $n$ nodes. We adopt linear Gaussian models as the latent causal model. We consider $N \in \{10^{2.5}, 10^3, 10^{3.5}, 10^4, 10^{4.5}, 10^5\}$ samples, and generate 100 latent models for each triplet $(N, n, d)$. We generate $N$ samples of $Z$ from each environment for each latent model. Transformation $\mathbf{G} \in \mathbb{R}^{d \times n}$ is randomly sampled under full-rank and bounded condition number constraints, and the observed variables are generated as $X = \mathbf{G} \cdot Z$. Due to the Gaussian structure of $Z$, the observed variables $X$ also have multivariate Gaussian distributions with

---

[2]The codebase for the experiments can be found at https://github.com/acarturk-e/finite-sample-linear-crl.

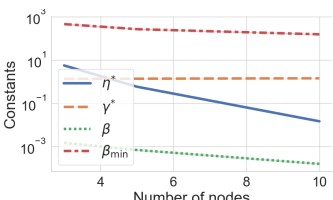
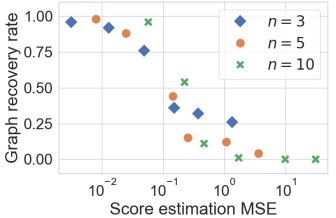

(a) Constants vs. problem dimension.  (b) Graph recovery vs. score MSE, $d = 15$

Figure 1: Numerical evaluations.

score function $\boldsymbol{s}_X(x) = -\Theta \cdot x$, where $\Theta$ is the precision matrix of $X$. Therefore, we estimate $\boldsymbol{s}_X$ using the sample precision matrix estimated from $N$ samples, $\hat{\Theta}_N$, as $\hat{\boldsymbol{s}}_X(x; \mathcal{X}_N) = -\hat{\Theta}_N \cdot x$.

**Results.** The sample complexity results depend on dimension $n$ and $d$ through their explicit presence as well as their implicit effect on model-dependent constants $\beta$, $\beta_{\min}$, $\eta^*$, and $\gamma^*$, as established in Theorems 1 and 2. First, we observe that all these parameters are independent of $d$. This is also expected theoretically since we can project down the $d$ dimensional observed space to the $n$ dimensional $\mathrm{col}(\mathbf{G})$. In Figure 1a, we illustrate the dependence of these constants on data dimensions $n$ and $d$. We observe that $\gamma^*$ remains constant, while $\beta$ and $\beta_{\min}$ are decreasing as $n$ increases, but the decay is not steep. On the other hand, we observe that $\eta^*$ decays exponentially with $n$. $\gamma^*$ is implicitly controlled by the condition number, which is kept bounded. We recall that $\beta_{\min}$ is the minimum among $\{\|\boldsymbol{d}_X^m\|_{p_X} : m \in [n]\}$, and similarly, $\beta$ is the inverse of the maximum of $\{\|\boldsymbol{d}_X^m\|_{p_X} : m \in [n]\}$, so they are expected to become smaller as $n$ increases. Finally, for $\eta^*$, we note that the definition in (22) is an order statistic among exponentially many, $2^n$, subsets.

Next, we note that all the imperfections in the graph and variable recovery are due to the imperfections in score estimates. In fact, this is the bottleneck in our decisions: the key impact of finite samples versus infinite samples is the degradation in the quality of score estimates. To have a direct insight into how the score estimation noise power translates into decision imperfection, we assess the quality of graph recovery success rate versus varying degrees of mean score error of score difference estimates $\mathbb{E}[\|\hat{\boldsymbol{d}}_X^m - \boldsymbol{d}_X^m\|_{p_X}^2]$ for different triplets $(N, n, d)$. In Figure 1b, each data point corresponds to a different sample size $N$, and the legend entries $\{3, 5, 10\}$ denote the latent dimension $n$. This figure illustrates the scatter plot of the error probability $\delta$ versus MSE for different parameter combinations and the corresponding local regression curves. It is observed that in the low MSE regime, $1 - \delta$ decays linearly with respect to $\log \mathbb{E}[\|\hat{\boldsymbol{d}}_X^m - \boldsymbol{d}_X^m\|_{p_X}^2]$ and it plateaus as the MSE increases. This trend is due to our algorithms' high sensitivity to errors in estimating approximate matrix ranks.

# 7 Conclusion

In this paper, we have established the first sample complexity results for interventional causal representation learning. Specifically, we have characterized upper bounds on the sample complexity of latent graph and variables recovery in terms of the finite sample performance of any consistent score difference estimator. We have, furthermore, adopted a particular score estimator [12] to derive explicit sample complexity statements.

Our sample complexity results are given for partial identifiability (up to ancestors) in the soft intervention and linear transformation setting. Establishing sample complexity results for perfect identifiability via hard interventions or considering general transformations are important future directions.

**Acknowledgements and disclosure of funding**

This work was supported by IBM through the IBM-Rensselaer Future of Computing Research Collaboration.

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

# Sample Complexity of Interventional Causal Representation Learning: Appendices

## Table of Contents

## A  Proof structure

We structure the proofs of our main results as follows.

- **Appendix B – Infinite sample guarantees:** First, we prove identifiability results in the noise-free setting, that is, using the ground truth $\mathbf{R}_X^m$ matrices instead of using the estimated $\hat{\mathbf{R}}_X^m$. These intermediate results lay the groundwork for the proofs for the finite sample setting.

- **Appendix C – Bounded noise guarantees:** Next, we show that if estimation error for the score difference functions, $\|\hat{\boldsymbol{d}}_X^m - \boldsymbol{d}_X^m\|_{p_X}$, is bounded, then the noise-free identifiability guarantees transfer to the noisy setting.

- **Appendix D – Sample complexity guarantees:** We use the noisy identifiability guarantees to prove that for any consistent score difference estimator, we can guarantee $(\epsilon, \delta)$–PAC guarantees for CRL identifiability objectives.

- **Appendix E – RKHS-based score estimator:** Finally, we use the finite sample guarantees provided by the RKHS-based score estimator of [12] to adapt the sample complexity statements for a generic estimator to explicit guarantees in terms of model-dependent constants.

To be used throughout the proof, we define the correlation matrices of $\boldsymbol{d}_Z^m$ as

$$\mathbf{R}_Z^m \triangleq \mathbb{E}_{p_Z}\left[\boldsymbol{d}_Z^m(z) \cdot (\boldsymbol{d}_Z^m(z))^\top\right] . \tag{34}$$

We recall the correlation matrices $\mathbf{R}_X^m$ and $\hat{\mathbf{R}}_X^m$ specified in (16). For a subset $\mathcal{M} \subseteq [n]$, we use the following shorthand notations for sums of the matrices in $\{\mathbf{R}_X^m : m \in \mathcal{M}\}$ and $\{\hat{\mathbf{R}}_X^m : m \in \mathcal{M}\}$.

$$\mathbf{R}_X^{\mathcal{M}} \triangleq \sum_{m \in \mathcal{M}} \mathbf{R}_X^m , \quad \text{and} \quad \hat{\mathbf{R}}_X^{\mathcal{M}} \triangleq \sum_{m \in \mathcal{M}} \hat{\mathbf{R}}_X^m , \qquad \forall \mathcal{M} \subseteq [n] . \tag{35}$$

# B  Identifiability guarantees in infinite sample regime

In this section, we provide the identifiability guarantees of Algorithms 1–3 with no error and probability 1 when using $\{\mathbf{R}_X^m \,:\, m \in [n]\}$ as input.

Denote the image of a function $f\colon \mathbb{R}^n \to \mathbb{R}^k$ for any $k \in \mathbb{Z}_+$ by $\mathsf{im}(f)$, that is,

$$\mathsf{im}(f) \triangleq \left\{ \, f(z) \,:\, z \in \mathbb{R}^n \, \right\} . \tag{36}$$

First, we note a property of the column space of correlation matrices.

**Lemma 6.** *For a continuous function $f\colon \mathbb{R}^n \to \mathbb{R}^k$, span of the image of $f$ equals to*

$$\mathsf{span}\,\mathsf{im}(f) = \mathsf{col}\left( \mathbb{E}_{p_Z}[f(z) \cdot f(z)^\top] \right) . \tag{37}$$

*Proof:* Note that two subspaces are equal if and only if their orthogonal complements are equal. We denote the orthogonal complement of $\mathsf{span}\,\mathsf{im}(f)$ by

$$\mathcal{S}_1 \triangleq \left( \mathsf{span}\,\mathsf{im}(f) \right)^\perp = \left\{ \, \mathbf{y} \in \mathbb{R}^k \,:\, \mathbf{y}^\top \cdot f(z) = 0 \quad \forall z \in \mathbb{R}^n \, \right\} . \tag{38}$$

The orthogonal complement of the column space of a symmetric matrix is the null space, which we denote by

$$\mathcal{S}_2 \triangleq \mathsf{null}\left( \mathbb{E}_{p_Z}[f(z) \cdot f(z)^\top] \right) = \left\{ \, \mathbf{y} \in \mathbb{R}^k \,:\, \mathbb{E}_{p_Z}[f(z) \cdot f(z)^\top] \cdot \mathbf{y} = 0 \, \right\} . \tag{39}$$

Since any correlation matrix is positive semidefinite, using Proposition 2, (39) can be written as

$$\mathcal{S}_2 = \left\{ \, \mathbf{y} \in \mathbb{R}^k \,:\, \mathbf{y}^\top \cdot \mathbb{E}_{p_Z}[f(z) \cdot f(z)^\top] \cdot \mathbf{y} = 0 \, \right\} \tag{40}$$

$$= \left\{ \, \mathbf{y} \in \mathbb{R}^k \,:\, \mathbb{E}_{p_Z}\left[ (\mathbf{y}^\top \cdot f(z))^2 \right] = 0 \, \right\} . \tag{41}$$

Since $f$ is a continuous function, so is $\mathbf{y}^\top \cdot f$ for any $\mathbf{y} \in \mathbb{R}^k$. We can then use [5, Proposition 2], which is stated below for the sake of conciseness.

**Proposition 1** ([5, Proposition 2]). *Consider two continuous functions $f, g : \mathbb{R}^n \to \mathbb{R}$. Then, for any $\alpha > 0$,*

$$\exists z \in \mathbb{R}^n \;\; f(z) \neq g(z) \quad \Longleftrightarrow \quad \mathbb{E}_{p_Z}\left[ \left| f(Z) - g(Z) \right|^\alpha \right] \neq 0 . \tag{42}$$

Under this proposition, (41) can be written as

$$\mathcal{S}_2 = \left\{ \, \mathbf{y} \in \mathbb{R}^k \,:\, \mathbf{y}^\top \cdot f(z) = 0 \quad \forall z \in \mathbb{R}^n \, \right\} , \tag{43}$$

which is the same as the definition of $\mathcal{S}_1$ in (38). This concludes the proof. ∎

**Corollary 1.** *For a continuous function $f\colon \mathbb{R}^d \to \mathbb{R}^k$, the span of the image of $f \circ \mathbf{G}$ equals to the column space of*

$$\mathbb{E}_{p_Z}[f(\mathbf{G} \cdot z) \cdot (f(\mathbf{G} \cdot z))^\top] = \mathbb{E}_{p_X}[f(x) \cdot (f(x))^\top] . \tag{44}$$

We can specialize this lemma for $\boldsymbol{d}_Z^m$ and $\boldsymbol{d}_X^m$ for any $m \in [n]$, which yields

$$\mathsf{col}(\mathbf{R}_Z^m) = \mathsf{span}\left\{ \, \boldsymbol{d}_Z^m(z) \,:\, z \in \mathbb{R}^n \, \right\} , \tag{45}$$

and

$$\mathsf{col}(\mathbf{R}_X^m) = \mathsf{span}\left\{ \, \boldsymbol{d}_X^m(x) \,:\, x \in \mathsf{col}(\mathbf{G}) \, \right\} . \tag{46}$$

## B.1  Proof of Lemma 2

We can state (17) from Lemma 1 equivalently as

$$\exists z \in \mathbb{R}^n \quad \left( \boldsymbol{d}_Z^m(z) \right)_i \neq 0 \iff i \in \overline{\mathrm{pa}}(I^m) . \tag{47}$$

Recall from (18) in Lemma 1 that $\boldsymbol{d}_X^m$ is related to $\boldsymbol{d}_Z^m$ by

$$\boldsymbol{d}_X^m(x) = (\mathbf{G}^\dagger)^\top \cdot \boldsymbol{d}_Z^m(z) = \sum_{i \in [n]} \left( \mathbf{G}^\dagger \right)_{i,:} \cdot \left( \boldsymbol{d}_Z^m(z) \right)_i , \quad x = \mathbf{G} \cdot z . \tag{48}$$

Then, due to (47), for any given $z \in \mathbb{R}^n$ and corresponding $x = \mathbf{G} \cdot z$, we have

$$\boldsymbol{d}_X^m(x) = \sum_{i \in \overline{\mathrm{pa}}(I^m)} \left(\mathbf{G}^\dagger\right)_{i,:} \cdot \left(\boldsymbol{d}_Z^m(z)\right)_i \in \mathsf{span}\left\{\left(\mathbf{G}^\dagger\right)_{i,:} \ : \ i \in \overline{\mathrm{pa}}(I^m)\right\} . \tag{49}$$

Since $\boldsymbol{d}_X^m(x)$ lies in the subspace $\mathsf{span}\{(\mathbf{G}^\dagger)_{i,:} \ : \ i \in \overline{\mathrm{pa}}(I^m)\}$ for any $x \in \mathsf{col}(\mathbf{G})$, we have

$$\mathsf{span}\left\{\boldsymbol{d}_X^m(x) \ : \ x \in \mathsf{col}(\mathbf{G})\right\} \subseteq \mathsf{span}\left\{\left(\mathbf{G}^\dagger\right)_{i,:} \ : \ i \in \overline{\mathrm{pa}}(I^m)\right\} . \tag{50}$$

Due to (46), we have

$$\mathsf{span}\left\{\boldsymbol{d}_X^m(x) \ : \ x \in \mathsf{col}(\mathbf{G})\right\} = \mathsf{col}(\mathbf{R}_X^m) , \tag{51}$$

which concludes the proof. ∎

## B.2 Encoder recovery

We can use Lemma 2 to show that in the infinite sample regime, Algorithm 3 works correctly. Specifically, let us show the following.

**Lemma 7** (Infinite samples – Encoder). *For any* $\mathbf{v} \in \mathsf{col}(\mathbf{R}_X^m)$, *we have*

$$\left(\mathbf{v}^\top \cdot \mathbf{G}\right)_i \neq 0 \implies i \in \overline{\mathrm{pa}}(I^m) . \tag{52}$$

*Proof:* Due to Lemma 2, for any $\mathbf{v} \in \mathsf{col}(\mathbf{R}_X^m)$, we have

$$\mathbf{v} \in \mathsf{span}\left\{\left(\mathbf{G}^\dagger\right)_{i,:} \ : \ i \in \overline{\mathrm{pa}}(I^m)\right\} . \tag{53}$$

In other words, any $\mathbf{v} \in \mathsf{col}(\mathbf{R}_X^m)$ can be expressed as

$$\mathbf{v} = \sum_{i \in \overline{\mathrm{pa}}(I^m)} c_i \cdot \left(\left(\mathbf{G}^\dagger\right)^\top\right)_{:,i} , \tag{54}$$

for some constants $\{c_i \ : \ i \in \overline{\mathrm{pa}}(I^m)\}$. Then, we have

$$\mathbf{v}^\top \cdot \mathbf{G} = \left(\sum_{i \in \overline{\mathrm{pa}}(I^m)} c_i \cdot \left(\mathbf{G}^\dagger\right)_{i,:}\right) \cdot \mathbf{G} = \sum_{i \in \overline{\mathrm{pa}}(I^m)} c_i \cdot \left(\mathbf{G}^\dagger\right)_{i,:} \cdot \mathbf{G} . \tag{55}$$

Since $\mathbf{G}$ has full column rank, the pseudoinverse $\mathbf{G}^\dagger$ satisfies $\mathbf{G}^\dagger \cdot \mathbf{G} = \mathbf{I}_n$, that is,

$$\left(\mathbf{G}^\dagger\right)_{i,:} \cdot \mathbf{G} = \mathbf{e}_i^\top , \tag{56}$$

where $\mathbf{e}_i$ denotes the $i$-th standard basis vector. Substituting this, (55) becomes

$$\mathbf{v}^\top \cdot \mathbf{G} = \sum_{i \in \overline{\mathrm{pa}}(I^m)} c_i \cdot \mathbf{e}_i^\top . \tag{57}$$

Note that $i$-th entry of a row vector $\mathbf{w} \in \mathbf{R}^n$ is given by $\mathbf{w} \cdot \mathbf{e}_i$. Therefore, we have

$$\left(\mathbf{v}^\top \cdot \mathbf{G}\right)_i = \begin{cases} c_i & i \in \overline{\mathrm{pa}}(I^m) , \\ 0 & \text{otherwise} . \end{cases} \tag{58}$$

This concludes the proof. ∎

## B.3 Equivalence of function rank to correlation rank

In this section, we adapt the workhorse lemmas of [5] to our setting.

**Lemma 8** ([5, Lemmas 5 and 6]). *Consider set* $\mathcal{A} \subseteq [n]$ *such that* $I^{\mathcal{A}} \triangleq \{I^m \ : \ m \in \mathcal{A}\}$ *is ancestrally closed, i.e.,* $\overline{\mathrm{pa}}(I^{\mathcal{A}}) = I^{\mathcal{A}}$. *Then,*

$$\mathsf{rank}\left(\mathbf{R}_X^{\mathcal{A}}\right) = |\mathcal{A}| . \tag{59}$$

*Also, under Assumption 1, for any* $k \in \mathcal{A}$, *we have*

$$\mathsf{rank}\left(\mathbf{R}_X^{\mathcal{A} \setminus \{k\}}\right) = \begin{cases} |\mathcal{A}| & \text{if } \exists j \in \mathcal{A} \setminus \{k\} \text{ such that } I^k \in \mathrm{pa}(I^j) , \\ |\mathcal{A}| - 1 & \text{otherwise} . \end{cases} \tag{60}$$

We first note that due to (18) and the definitions of $\mathbf{R}_X^m$ and $\mathbf{R}_Z^m$ in (16) and (34), we have

$$\mathbf{R}_X^m = (\mathbf{G}^\dagger)^\top \cdot \mathbf{R}_Z^m \cdot \mathbf{G}^\dagger \,, \tag{61}$$

where $\mathbf{G}^\dagger$ is a full row rank matrix. Therefore, for any $\mathcal{B} \subseteq [n]$,

$$\mathsf{rank}(\mathbf{R}_X^\mathcal{B}) = \mathsf{rank}(\mathbf{R}_Z^\mathcal{B}) \,. \tag{62}$$

Next, we note that for a function $f \colon \mathbb{R}^n \to \mathbb{R}^k$, the notation $\mathcal{R}(f)$ in [5] is used to denote the "rank" of a function, and it corresponds to $\dim \mathsf{span}\,\mathsf{im}(f)$ in the notation of this paper. Using Proposition 3 and Lemma 6, we can see that the notation $\mathcal{R}(\mathcal{F})$ for a set of functions $\mathcal{F}$ can be given as

$$\mathcal{R}(\mathcal{F}) = k - \dim(\bigcap_{f \in \mathcal{F}} (\mathsf{span}\,\mathsf{im}(f))^\perp) \tag{63}$$

$$= k - \dim(\bigcap_{f \in \mathcal{F}} \mathsf{null}(\mathbb{E}_{p_Z}[f(z) \cdot f(z)^\top])) \tag{64}$$

$$= k - \dim\mathsf{null}(\sum_{f \in \mathcal{F}} \mathbb{E}_{p_Z}[f(z) \cdot f(z)^\top]) \tag{65}$$

$$= \dim\mathsf{col}(\sum_{f \in \mathcal{F}} \mathbb{E}_{p_Z}[f(z) \cdot f(z)^\top]) \,. \tag{66}$$

Using (66) and (62), we can adopt [5, Lemmas 5 and 6] using $\{\mathbf{R}_X^m \,:\, m \in [n]\}$ matrices.

We note that (59) and (60) together with Lemma 2 imply that, for any ancestrally closed $\mathcal{A} \subseteq [n]$ and $k \in \mathcal{A}$ for which there exists $j \in \mathcal{A} \setminus \{k\}$ such that $I^k \in \mathsf{pa}(I^j)$, we have

$$\mathsf{col}\left(\mathbf{R}_X^\mathcal{A}\right) = \mathsf{col}\left(\mathbf{R}_X^{\mathcal{A} \setminus \{k\}}\right) = \mathsf{span}\left\{(\mathbf{G}^\dagger)^\top_{:,i} \,:\, i \in \overline{\mathsf{pa}}(I^\mathcal{A})\right\} \,. \tag{67}$$

We note that due to the property $\mathbf{G}^\dagger \cdot \mathbf{G} = \mathbf{I}_n$, we have, for any $\mathcal{M} \subseteq [n]$,

$$\mathsf{null}\left\{(\mathbf{G}^\dagger)^\top_{:,i} \,:\, i \in \mathcal{M}\right\} = \mathsf{span}\left\{\mathbf{G}_{:,i} \,:\, i \notin \mathcal{M}\right\}, \tag{68}$$

and

$$\mathsf{null}\left\{\mathbf{G}_{:,i} \,:\, i \in \mathcal{M}\right\} = \mathsf{span}\left\{(\mathbf{G}^\dagger)^\top_{:,i} \,:\, i \notin \mathcal{M}\right\}. \tag{69}$$

### B.4  Causal order recovery

We will show that Algorithm 1 gives a valid causal order in the infinite sample regime using Lemma 8.

**Lemma 9** (Infinite samples – Causal order). *Setting $\eta = 0$, $I \circ \pi$ is a valid causal order, where $\pi$ is the output of Algorithm 1 using $\{\mathbf{R}_X^m \,:\, m \in [n]\}$.*

*Proof:*  The proof of this lemma follows closely to the proof of [5, Lemma 7]. Note that setting $\eta = 0$ means that approximate rank and column/null spaces are actually equal to the standard rank and column/null space definitions. We note that $I \circ \pi$ is a valid causal order if and only if, for any $t \in [n]$, we have

$$\mathsf{de}(I^{\pi_t}) \subseteq \left\{I^{\pi_i} \,:\, i > t\right\} \,. \tag{70}$$

We prove (70) by induction as follows.

In the base case, for $t = n$, we have $\mathcal{V}_t = [n]$. Therefore, using Lemma 8, we see that

$$\mathsf{rank}(\mathbf{R}_X^{\mathcal{V}_t \setminus \{k\}}) = \begin{cases} n & \mathsf{de}(I^k) \cap I^{\mathcal{V}_t} \neq \emptyset \,, \\ n-1 & \mathsf{de}(I^k) \cap I^{\mathcal{V}_t} = \emptyset \,. \end{cases} \tag{71}$$

Therefore, for $t = n$, $\mathsf{rank}(\mathbf{R}_X^{\mathcal{V}_t \setminus \{k\}}) = t - 1$ in Algorithm 1 if and only if $k \in \mathcal{V}_t$ satisfies

$$\emptyset = \mathsf{de}(I^k) \cap I^{\mathcal{V}_t} = \mathsf{de}(I^k) \cap [n] = \mathsf{de}(I^k) \,. \tag{72}$$

Since $\left\{I^{\pi_i} \,:\, i > t\right\} = \emptyset$ for $t = n$, the base case is proved by choosing any such $k$ as $\pi_t$.

For the induction step at time $t \in \{n-1, \ldots, 1\}$, we assume that, for any $u \in \{t+1, \ldots, n\}$, we have $\mathsf{de}(I^{\pi_u}) \subseteq \{I^{\pi_i} \,:\, i > u\}$. By construction, $\mathcal{V}_t = [n] \setminus \{\pi_u \,:\, u > t\}$. Therefore, $I^{\mathcal{V}_t}$ is

ancestrally closed: For any $k \in I^{\mathcal{V}_t}$, $k$ cannot be a descendant of $I^{[n] \backslash \mathcal{V}_t} = \{ I^{\pi_u} : u > t \}$ due to the induction hypothesis. In other words, $\overline{\mathrm{pa}}(I^{\mathcal{V}_t}) = I^{\mathcal{V}_t}$. Therefore, we can use Lemma 8 as

$$\mathsf{rank}(\mathbf{R}_X^{\mathcal{V}_t \backslash \{k\}}) = \begin{cases} t & \mathrm{de}(I^k) \cap I^{\mathcal{V}_t} \neq \emptyset \,, \\ t - 1 & \mathrm{de}(I^k) \cap I^{\mathcal{V}_t} = \emptyset \,. \end{cases} \tag{73}$$

Therefore, for any $t \in \{ n-1, \dots, 1 \}$, $\mathsf{rank}(\mathbf{R}_X^{\mathcal{V}_t \backslash \{k\}}) = t - 1$ in Algorithm 1 if and only if $k \in \mathcal{V}_t$ satisfies

$$\emptyset = \mathrm{de}(I^k) \cap I^{\mathcal{V}_t} \,. \tag{74}$$

This is equivalent to

$$\mathrm{de}(I^k) \subseteq [n] \setminus I^{\mathcal{V}_t} = I^{[n] \backslash \mathcal{V}_t} = \{ I^{\pi_u} : u > t \} \,. \tag{75}$$

Therefore, choosing any such $k$ as $\pi_t$ satisfies the induction hypothesis for $t$ as well, and the proof by induction is concluded. ∎

## B.5 Graph recovery

**Lemma 10** (Infinite samples – Graph). *Setting $\eta = \gamma = 0$, the collective output of Algorithms 1 and 2 using $\{ \mathbf{R}_X^m : m \in [n] \}$ is equal to the graph isomorphism of the transitive closure of $\mathcal{G}$ by permutation $I$.*

*Proof:* The proof of this lemma follows closely to the proof of [5, Lemma 8]. Note that setting $\eta = 0$ means that approximate ranks and column/null spaces are equal to the standard definitions. Similarly, $\gamma = 0$ means that approximate orthogonality is equivalent to the standard orthogonality definition. We first note that the transitive closure of two DAGs are equal if and only if their transitive *reductions* are equal. Therefore, the lemma statement is equivalent to

$$\pi_t \in \hat{\mathrm{pa}}_{\mathrm{tr}}(\pi_j) \iff I^{\pi_t} \in \mathrm{pa}_{\mathrm{tr}}(I^{\pi_j}) \,, \qquad \forall t, j \in [n] \,. \tag{76}$$

We will prove this result by induction over the transitive reduction edges. In the rest of the proof, using Lemma 9, we will leverage that $I \circ \pi$ is a valid causal order.

In the base case, we have $t = n - 1$, and thus necessarily $j = n$. The only possible path between $I^{\pi_t}$ to $I^{\pi_j}$ is therefore a transitive reduction edge. Since we did not add any children to node $\pi_t$ yet, the set $\mathcal{M}_{t,j}$ is equal to $\mathcal{V}_j \setminus \{\pi_t\}$. Noting that $I^{\mathcal{V}_j}$ is ancestrally closed and the only possible child of $I^{\pi_t}$ is $I^{\pi_j}$ when $I \circ \pi$ is a valid causal order, we can use Lemma 8 to obtain

$$\mathsf{rank}(\mathbf{R}_X^{\mathcal{M}_{t,j}}) = \begin{cases} n & I^{\pi_j} \in \mathrm{ch}(I^{\pi_t}) \,, \\ n - 1 & I^{\pi_j} \notin \mathrm{ch}(I^{\pi_t}) \,. \end{cases} \tag{77}$$

Specifically, using (67), we can specify the exact column spaces we can obtain from $\mathcal{M}_{t,j}$ as

$$\mathsf{col}(\mathbf{R}_X^{\mathcal{M}_{t,j}}) = \begin{cases} \mathsf{span} \{ (\mathbf{G}^\dagger)^\top_{:,i} : i \in I^{\mathcal{V}_j} \} & I^{\pi_j} \in \mathrm{ch}(I^{\pi_t}) \,, \\ \mathsf{span} \{ (\mathbf{G}^\dagger)^\top_{:,i} : i \in I^{\mathcal{V}_j \backslash \{\pi_t\}} \} & I^{\pi_j} \notin \mathrm{ch}(I^{\pi_t}) \,. \end{cases} \tag{78}$$

Then, using (68), the null space of $\mathbf{R}_X^{\mathcal{M}_{t,j}}$ is given by

$$\mathsf{null}(\mathbf{R}_X^{\mathcal{M}_{t,j}}) = \begin{cases} \mathsf{span} \{ \mathbf{G}_{:,i} : i \notin I^{\mathcal{V}_j} \} & I^{\pi_j} \in \mathrm{ch}(I^{\pi_t}) \,, \\ \mathsf{span} \{ \mathbf{G}_{:,i} : i \notin I^{\mathcal{V}_j \backslash \{\pi_t\}} \} & I^{\pi_j} \notin \mathrm{ch}(I^{\pi_t}) \,. \end{cases} \tag{79}$$

The test in Algorithm 2 involves checking whether the column space of $\mathbf{R}_X^{\mathcal{V}_t}$ and the null space of $\mathbf{R}_X^{\mathcal{M}_{t,j}}$ are orthogonal. Note that $I^{\mathcal{V}_t}$ is ancestrally closed, therefore, using (67), we get

$$\mathsf{col}(\mathbf{R}_X^{\mathcal{V}_t}) = \mathsf{span} \{ (\mathbf{G}^\dagger)^\top_{:,i} : i \in \overline{\mathrm{pa}}(I^{\mathcal{V}_t}) \} = \mathsf{span} \{ (\mathbf{G}^\dagger)^\top_{:,i} : i \in I^{\mathcal{V}_t} \} \,. \tag{80}$$

Next, we note that the orthogonality test which uses orthonormal bases of subspaces (Definition 5) is equal to the following. Let $\mathbf{A} \in \mathbb{R}^{d \times k_1}$ and $\mathbf{B} \in \mathbb{R}^{d \times k_2}$ be full column rank matrices. Then,

$$\mathsf{col}(\mathbf{A}) \perp \mathsf{col}(\mathbf{B}) \iff \left\| \mathbf{A} \cdot \mathbf{A}^\dagger \cdot \mathbf{B} \cdot \mathbf{B}^\dagger \right\|_2 = 0 \,. \tag{81}$$

This holds since $\mathbf{A} \cdot \mathbf{A}^\dagger$ is the orthogonal projection operator onto $\mathrm{col}(\mathbf{A})$. Therefore, given orthonormal bases $\mathbf{A}_0$ and $\mathbf{B}_0$ for $\mathbf{A}$ and $\mathbf{B}$, we have $\mathbf{A} \cdot \mathbf{A}^\dagger = \mathbf{A}_0 \cdot \mathbf{A}_0^\top$ and $\mathbf{B} \cdot \mathbf{B}^\dagger = \mathbf{B}_0 \cdot \mathbf{B}_0^\top$. Then,

$$\left\| \mathbf{A} \cdot \mathbf{A}^\dagger \cdot \mathbf{B} \cdot \mathbf{B}^\dagger \right\|_2 = \left\| \mathbf{A}_0 \cdot \mathbf{A}_0^\top \cdot \mathbf{B}_0 \cdot \mathbf{B}_0^\top \right\|_2 = \left\| \mathbf{A}_0^\top \cdot \mathbf{B}_0 \right\|_2, \tag{82}$$

which is precisely the metric used in Definition 5. To use this property, let us first consider the matrix $\mathbf{G}_{:,\mathcal{A}}$ which is formed by stacking the independent column vectors in set $\{\mathbf{G}_{:,i} \,:\, i \in \mathcal{A}\}$ for some subset $\mathcal{A} \subseteq [n]$. Since $\mathbf{G}$ has full column rank, we see that the pseudoinverse of $\mathbf{G}_{:,\mathcal{A}}$ is exactly given by

$$\left( \mathbf{G}_{:,\mathcal{A}} \right)^\dagger = \left( \mathbf{G}^\dagger \right)_{\mathcal{A},:}, \tag{83}$$

where the matrix $\left( \mathbf{G}^\dagger \right)_{\mathcal{A},:}$ is formed by stacking the independent row vectors in set $\{ (\mathbf{G}^\dagger)_{i,:} \,:\, i \in \mathcal{A} \}$. Using this notation, if we have

$$\mathsf{col}(\mathbf{R}_X^{\mathcal{M}_{t,j}}) = \mathsf{span} \left\{ (\mathbf{G}^\dagger)^\top_{:,i} \,:\, i \in \overline{\mathrm{pa}}(I^{\mathcal{M}_{t,j}}) \right\}, \tag{84}$$

then the orthogonality metric is equal to

$$\left\| (\mathbf{G}^\dagger)^\top_{:,I^{\mathcal{V}_t}} \cdot \mathbf{G}^\top_{I^{\mathcal{V}_t},:} \cdot \mathbf{G}_{:,I^{[n]\setminus \overline{\mathrm{pa}}(\mathcal{M}_{t,j})}} \cdot \mathbf{G}^\dagger_{I^{[n]\setminus \overline{\mathrm{pa}}(\mathcal{M}_{t,j})},:} \right\|_2 \tag{85}$$

$$= \left\| \mathbf{G}_{:,I^{\mathcal{V}_t}} \cdot \mathbf{G}^\dagger_{I^{\mathcal{V}_t},:} \cdot \mathbf{G}_{:,I^{[n]\setminus \overline{\mathrm{pa}}(\mathcal{M}_{t,j})}} \cdot \mathbf{G}^\dagger_{I^{[n]\setminus \overline{\mathrm{pa}}(\mathcal{M}_{t,j})},:} \right\|_2 \tag{86}$$

$$= \left\| \mathbf{G}_{:,I^{\mathcal{V}_t}} \cdot \mathbf{I}_{I^{\mathcal{V}_t},I^{[n]\setminus \overline{\mathrm{pa}}(\mathcal{M}_{t,j})}} \cdot \mathbf{G}^\dagger_{I^{[n]\setminus \overline{\mathrm{pa}}(\mathcal{M}_{t,j})},:} \right\|_2 \tag{87}$$

$$= \left\| \sum_{i \in I^{\mathcal{V}_t}\setminus \overline{\mathrm{pa}}(I^{\mathcal{M}_{t,j}})} \mathbf{G}_{:,i} \cdot \mathbf{G}^\dagger_{i,:} \right\|_2. \tag{88}$$

Due to the construction of sets $\mathcal{V}_t$ and $\mathcal{M}_{t,j}$, we have $I^{\mathcal{V}_t} \setminus I^{\mathcal{M}_{t,j}} = I^{\pi_t}$. Therefore, the only entry in $I^{\mathcal{V}_t}$ that may be missing from $\overline{\mathrm{pa}}(I^{\mathcal{M}_{t,j}})$ is $I^{\pi_t}$. Therefore, previous equation is equal to

$$\left\| (\mathbf{G}^\dagger)^\top_{:,I^{\mathcal{V}_t}} \cdot \mathbf{G}^\top_{I^{\mathcal{V}_t},:} \cdot \mathbf{G}_{:,I^{[n]\setminus \overline{\mathrm{pa}}(\mathcal{M}_{t,j})}} \cdot \mathbf{G}^\dagger_{I^{[n]\setminus \overline{\mathrm{pa}}(\mathcal{M}_{t,j})},:} \right\|_2 \tag{89}$$

$$= \begin{cases} \left\| \mathbf{G}_{:,I^{\pi_t}} \right\|_2 \cdot \left\| \mathbf{G}^\dagger_{I^{\pi_t},:} \right\|_2 & I^{\pi_t} \notin \mathrm{pa}(I^{\mathcal{M}_{t,j}}), \\ 0 & I^{\pi_t} \in \mathrm{pa}(I^{\mathcal{M}_{t,j}}). \end{cases} \tag{90}$$

In summary, if (84) holds, the orthogonality inspected in Algorithm 2 happens if and only if

$$I^{\pi_t} \in \mathrm{pa}(I^{\mathcal{M}_{t,j}}). \tag{91}$$

In the base case, since $t = n - 1$, this holds if and only if $I^{\pi_t} \in \mathrm{pa}_{\mathrm{tr}}(I^{\pi_j})$, and the base case is done.

For the induction hypothesis, we assume that for any $u \in \{n-1,\dots,t+1\}$, all the transitive reductive edges has been correctly identified, that is, for any $i \in [n]$, $\pi_u \in \hat{\mathrm{pa}}_{\mathrm{tr}}(\pi_i)$ if and only if $I^{\pi_u} \in \mathrm{pa}_{\mathrm{tr}}(I^{\pi_i})$. We show that the induction hypothesis is also satisfied for $t$ and $j \in [n]$ using induction on $j$.

In the base case, we have $j = t + 1$. This case works exactly the same way as the $t = n-1$ and $j = n$ case. For the induction hypothesis, assume that for any $u \in \{n+1,\dots,j-1\}$, all the transitive reductive edges between $I^{\pi_t}$ and $I^{\pi_u}$ have been correctly identified, that is, $\pi_t \in \hat{\mathrm{pa}}_{\mathrm{tr}}(\pi_u)$ if and only if $I^{\pi_t} \in \mathrm{pa}_{\mathrm{tr}}(I^{\pi_u})$. We prove that the hypothesis also holds for $j$ as follows. First, we claim that $I^{\mathcal{M}_{t,j}\cup\{\pi_t\}}$ is an ancestrally closed set. Note that $I^{\mathcal{V}_j}$ is an ancestrally closed set by construction. Since the induction hypothesis ensures that all transitive reduction relations between $I^{\pi_u}$ and $I^{\pi_i}$ for any $u \in \{t+1,\dots,n-1\}$ and $i \in [n]$ are recovered, this also means that transitive closure relations, that is, descendants, are also recovered. Namely, the induction hypothesis implies

$$I^{\pi_i} \in \mathrm{de}(I^{\pi_u}) \iff \pi_i \in \hat{\mathrm{de}}(\pi_u), \quad \forall u \in \{t+1,\dots,n-1\}, \forall i \in [n]. \tag{92}$$

Next, the induction hypothesis for $j$ implies that for any $k \in \{t+1,\dots,j-1\}$, we have also identified all the transitive reductive edges $I^{\pi_t} \to I^{\pi_k}$. Therefore, for set $\mathcal{M}_{t,j} \cup \{\pi_t\} = \mathcal{V}_j \setminus \hat{\mathrm{ch}}(\pi_t)$, all the elements in $\hat{\mathrm{ch}}(\pi_t)$ has the property that their descendants have been fully identified, and therefore are excluded from the set ($\hat{\mathcal{G}}$ keeps the transitive closure), which means the set $I^{\mathcal{M}_{t,j}\cup\{\pi_t\}}$ is ancestrally closed. Since this set is ancestrally closed, we have

$$\overline{\mathrm{pa}}(I^{\mathcal{M}_{t,j}}) = \begin{cases} I^{\mathcal{M}_{t,j}} & I^{\pi_t} \notin \mathrm{pa}(I^{\pi_j}), \\ I^{\mathcal{M}_{t,j}} \cup \{\pi_t\} & I^{\pi_t} \in \mathrm{pa}(I^{\pi_j}), \end{cases} \tag{93}$$

or, using Lemma 8,

$$\operatorname{col}(\mathbf{R}_X^{\mathcal{M}_{t,j}}) = \operatorname{span}\left\{ (\mathbf{G}^\dagger)^\top_{:,i} \,:\, i \in \overline{\operatorname{pa}}(I^{\mathcal{M}_{t,j}}) \right\}. \tag{94}$$

In the base case $t = n$, we had shown that this was a sufficient condition for ensuring that Algorithm 2 detects the transitive reductive edges correctly. Therefore, we can detect any existing transitive reduction edges between $I^{\pi_t}$ and $I^{\pi_j}$ for any $j$ and $t$, which concludes the proof. ∎

## C  Identifiability guarantees under bounded score estimation noise

In this section, we show that as long as the error in the estimate of the score differences, $\max_{m \in [n]} \|\hat{d}_X^m - d_X^m\|_{p_X}$, is upper bounded by a model-dependent constant, the results in the infinite data regime still hold.

The first key observation is that the error $\|\hat{\mathbf{R}}_X^m - \mathbf{R}_X^m\|_2$ is bounded in terms of $\|\hat{d}_X^m - d_X^m\|_{p_X}$.

**Lemma 11.** *In high signal-to-noise ratio regime for score differences, that is, when*

$$\left\|\hat{d}_X^m - d_X^m\right\|_{p_X} \le 2 \cdot \left\|d_X^m\right\|_{p_X}, \tag{95}$$

*the error in correlation matrix $\mathbf{R}_X^m$ is bounded by the error in $d_X^m$ as*

$$\left\|\hat{\mathbf{R}}_X^m - \mathbf{R}_X^m\right\|_2 \le 4 \cdot \left\|d_X^m\right\|_{p_X} \cdot \left\|\hat{d}_X^m - d_X^m\right\|_{p_X}. \tag{96}$$

*Proof:* First, note that

$$\hat{\mathbf{R}}_X^m - \mathbf{R}_X^m = \mathbb{E}_{p_X}\left[\hat{d}_X^m(x) \cdot \hat{d}_X^m(x)^\top - d_X^m(x) \cdot d_X^m(x)^\top\right] \tag{97}$$

$$= \mathbb{E}_{p_X}\left[(\hat{d}_X^m(x) - d_X^m(x)) \cdot (\hat{d}_X^m(x) - d_X^m(x))^\top \right.$$
$$+ d_X^m(x) \cdot (\hat{d}_X^m(x) - d_X^m(x))^\top \tag{98}$$
$$\left. + (\hat{d}_X^m(x) - d_X^m(x)) \cdot d_X^m(x)^\top\right].$$

Next, we apply the spectral norm.

$$\left\|\hat{\mathbf{R}}_X^m - \mathbf{R}_X^m\right\|_2 = \left\|\mathbb{E}_{p_X}\left[(\hat{d}_X^m(x) - d_X^m(x)) \cdot (\hat{d}_X^m(x) - d_X^m(x))^\top\right.\right.$$
$$+ d_X^m(x) \cdot (\hat{d}_X^m(x) - d_X^m(x))^\top \tag{99}$$
$$\left.\left. + (\hat{d}_X^m(x) - d_X^m(x)) \cdot d_X^m(x)^\top\right]\right\|_2$$
$$\le \left\|\mathbb{E}_{p_X}\left[(\hat{d}_X^m(x) - d_X^m(x)) \cdot (\hat{d}_X^m(x) - d_X^m(x))^\top\right]\right\|_2$$
$$+ 2 \cdot \left\|\mathbb{E}_{p_X}\left[d_X^m(x) \cdot (\hat{d}_X^m(x) - d_X^m(x))^\top\right]\right\|_2. \tag{100}$$

Note that the first term is the spectral norm of a covariance matrix, and is upper bounded by

$$\left\|\mathbb{E}_{p_X}\left[(\hat{d}_X^m(x) - d_X^m(x)) \cdot (\hat{d}_X^m(x) - d_X^m(x))^\top\right]\right\|_2 \le \mathbb{E}_{p_X}\left\|\hat{d}_X^m(x) - d_X^m(x)\right\|_2^2 \tag{101}$$
$$= \left\|\hat{d}_X^m - d_X^m\right\|_{p_X}^2. \tag{102}$$

For the second term, we can upper bound the spectral norm by Frobenius norm as

$$\left\|\mathbb{E}_{p_X}\left[d_X^m(x) \cdot (\hat{d}_X^m(x) - d_X^m(x))^\top\right]\right\|_2^2 \le \left\|\mathbb{E}_{p_X}\left[d_X^m(x) \cdot (\hat{d}_X^m(x) - d_X^m(x))^\top\right]\right\|_F^2 \tag{103}$$
$$= \sum_{i,j=1}^d \left(\mathbb{E}_{p_X}\left[(d_X^m(x))_i \cdot (\hat{d}_X^m(x) - d_X^m(x))_j\right]\right)^2. \tag{104}$$

Using Cauchy–Schwarz inequality,

$$\left\|\mathbb{E}_{p_X}\left[d_X^m(x) \cdot (\hat{d}_X^m(x) - d_X^m(x))^\top\right]\right\|_2^2 \le \sum_{i,j=1}^d \mathbb{E}_{p_X}\left[(d_X^m(x))_i^2\right] \cdot \mathbb{E}_{p_X}\left[(\hat{d}_X^m(x) - d_X^m(x))_j^2\right] \tag{105}$$
$$= \mathbb{E}_{p_X}\left\|d_X^m(x)\right\|_2^2 \cdot \mathbb{E}_{p_X}\left\|\hat{d}_X^m(x) - d_X^m(x)\right\|_2^2 \tag{106}$$
$$= \left\|d_X^m\right\|_{p_X}^2 \cdot \left\|\hat{d}_X^m - d_X^m\right\|_{p_X}^2. \tag{107}$$

Therefore, we can upper bound $\|\hat{\mathbf{R}}_X^m - \mathbf{R}_X^m\|_2$ by

$$\left\|\hat{\mathbf{R}}_X^m - \mathbf{R}_X^m\right\|_2 \leq \left\|\hat{\boldsymbol{d}}_X^m - \boldsymbol{d}_X^m\right\|_{p_X}^2 + 2 \cdot \left\|\boldsymbol{d}_X^m\right\|_{p_X} \cdot \left\|\hat{\boldsymbol{d}}_X^m - \boldsymbol{d}_X^m\right\|_{p_X} . \tag{108}$$

Finally, we note that if (95) holds, then this bound simplifies to (96). ∎

Our finite sample CRL algorithms use approximate column and null spaces of (sums of) noisy score difference correlation matrices $\hat{\mathbf{R}}_X^{\mathcal{M}}$, where $\mathcal{M} \subseteq [n]$. Therefore, we need to ensure that these approximate subspaces (i) have the same rank as, and (ii) are close to their noise-free counterparts.

First, we show that under bounded score difference estimation noise, the approximate rank of $\hat{\mathbf{R}}_X^{\mathcal{M}}$ is equal to the rank of $\mathbf{R}_X^{\mathcal{M}}$. To show this, we will leverage Weyl's inequality.

**Lemma 12** (Weyl's inequality). *For any two symmetric $k \times k$ real matrices $\mathbf{A}, \mathbf{B}$, we have*

$$\left\|\boldsymbol{\lambda}(\mathbf{A}) - \boldsymbol{\lambda}(\mathbf{B})\right\|_\infty \leq \left\|\mathbf{A} - \mathbf{B}\right\|_2 , \tag{109}$$

*where $\boldsymbol{\lambda}(\mathbf{A})$ for symmetric matrix $\mathbf{A}$ denotes the vector of eigenvalues of $\mathbf{A}$ ordered in ascending order, and $\|\mathbf{v}\|_\infty$ is the maximum absolute value among the entries of $\mathbf{v}$.*

We can now show that the approximate rank of $\hat{\mathbf{R}}_X^m$ matrices and sums are equal to the rank of $\mathbf{R}_X^m$ under proper threshold selection and low error.

**Lemma 13.** *Define $\eta^*$ as the minimum nonzero eigenvalue among $\{\mathbf{R}_X^{\mathcal{M}} : \mathcal{M} \subseteq [n]\}$, that is,*

$$\eta^* \triangleq \min_{\mathcal{M} \subseteq [n]} \min \left\{ \boldsymbol{\lambda}_i\left(\mathbf{R}_X^{\mathcal{M}}\right) : i \in [d], \, \boldsymbol{\lambda}_i\left(\mathbf{R}_X^{\mathcal{M}}\right) \neq 0 \right\} . \tag{110}$$

*(i) For $\eta \in (0, \eta^*)$, if*

$$\max_{m \in [n]} \left\|\hat{\mathbf{R}}_X^m - \mathbf{R}_X^m\right\|_2 < \min\{\eta, \eta^* - \eta\} = \eta_* , \tag{111}$$

*then, for any $m \in [n]$, we have*

$$\mathsf{dim\,col}\left(\mathbf{R}_X^m\right) = \mathsf{dim\,col}\left(\hat{\mathbf{R}}_X^m; \eta\right) . \tag{112}$$

*(ii) For $\eta \in (0, \eta^*)$, if*

$$\max_{m \in [n]} \left\|\hat{\mathbf{R}}_X^m - \mathbf{R}_X^m\right\|_2 < \frac{\eta_*}{n} , \tag{113}$$

*then, for any $\mathcal{M} \subseteq [n]$, we have*

$$\mathsf{dim\,col}\left(\mathbf{R}_X^{\mathcal{M}}\right) = \mathsf{dim\,col}\left(\hat{\mathbf{R}}_X^{\mathcal{M}}; \eta\right) . \tag{114}$$

*Proof:* Let us investigate the approximate rank of $\hat{\mathbf{R}}_X^{\mathcal{M}}$ under threshold $\eta$ for any $\mathcal{M} \subseteq [n]$. If the eigenvalues of $\hat{\mathbf{R}}_X^{\mathcal{M}}$ that correspond to the eigenvalues of the null space of $\mathbf{R}_X^{\mathcal{M}}$ are below the threshold $\eta$, and the eigenvalues of $\hat{\mathbf{R}}_X^{\mathcal{M}}$ that correspond to the eigenvalues of the column space of $\mathbf{R}_X^{\mathcal{M}}$ are above $\eta$, then the approximate rank of $\hat{\mathbf{R}}_X^{\mathcal{M}}$ is equal to the rank of $\mathbf{R}_X^{\mathcal{M}}$. Let us denote the maximum amount any eigenvalue of $\hat{\mathbf{R}}_X^{\mathcal{M}}$ differs from the corresponding eigenvalue of $\mathbf{R}_X^{\mathcal{M}}$ by

$$\chi \triangleq \left\|\boldsymbol{\lambda}(\hat{\mathbf{R}}_X^{\mathcal{M}}) - \boldsymbol{\lambda}(\mathbf{R}_X^{\mathcal{M}})\right\|_\infty . \tag{115}$$

Thus, the eigenvalues of the noisy correlation matrix $\hat{\mathbf{R}}_X^{\mathcal{M}}$ that correspond to the null space eigenvalues of $\mathbf{R}_X^{\mathcal{M}}$ is at most $\chi$. Similarly, eigenvalues of the noisy correlation matrix $\hat{\mathbf{R}}_X^{\mathcal{M}}$ that correspond to the column space eigenvalues of $\mathbf{R}_X^{\mathcal{M}}$ is at least

$$\min \left\{ \boldsymbol{\lambda}_i\left(\mathbf{R}_X^{\mathcal{M}}\right) : i \in [d], \, \boldsymbol{\lambda}_i\left(\mathbf{R}_X^{\mathcal{M}}\right) \neq 0 \right\} - \chi . \tag{116}$$

Therefore, if

$$\chi < \eta < \min \left\{ \boldsymbol{\lambda}_i\left(\mathbf{R}_X^{\mathcal{M}}\right) : i \in [d], \, \boldsymbol{\lambda}_i\left(\mathbf{R}_X^{\mathcal{M}}\right) \neq 0 \right\} - \chi , \tag{117}$$

then using threshold $\eta$ ensures that the approximate rank of $\hat{\mathbf{R}}_X^{\mathcal{M}}$ is equal to the rank of $\mathbf{R}_X^{\mathcal{M}}$. We can generalize this statement as follows. First, note that, by definition,

$$\eta^* \leq \min \left\{ \boldsymbol{\lambda}_i\left(\mathbf{R}_X^{\mathcal{M}}\right) : i \in [d], \, \boldsymbol{\lambda}_i\left(\mathbf{R}_X^{\mathcal{M}}\right) \neq 0 \right\} . \tag{118}$$

Then, the rank statement about $\hat{\mathbf{R}}_X^{\mathcal{M}}$ is correct if $\chi$ satisfies

$$\chi < \eta < \eta^* - \chi \, . \tag{119}$$

This statement can be equivalently written as

$$\chi < \min\{\eta, \eta^* - \eta\} \triangleq \eta_* \, . \tag{120}$$

Next, we find an upper bound on $\chi$ in terms of $\|\mathbf{R}_X^m - \mathbf{R}_X^m\|_2$ using Weyl's inequality (Lemma 12).

$$\chi = \left\| \boldsymbol{\lambda}(\hat{\mathbf{R}}_X^{\mathcal{M}}) - \boldsymbol{\lambda}(\mathbf{R}_X^{\mathcal{M}}) \right\|_\infty \leq \left\| \hat{\mathbf{R}}_X^{\mathcal{M}} - \mathbf{R}_X^{\mathcal{M}} \right\|_2 \tag{121}$$

$$\leq \sum_{m \in \mathcal{M}} \left\| \hat{\mathbf{R}}_X^m - \mathbf{R}_X^m \right\|_2 \tag{122}$$

$$\leq |\mathcal{M}| \cdot \max_{m \in [n]} \left\| \hat{\mathbf{R}}_X^m - \mathbf{R}_X^m \right\|_2 \tag{123}$$

$$\leq n \cdot \max_{m \in [n]} \left\| \hat{\mathbf{R}}_X^m - \mathbf{R}_X^m \right\|_2 \, . \tag{124}$$

Therefore, for any $\eta \in (0, \eta^*)$, leveraging the condition (120), we have

(i) If $|\mathcal{M}| = 1$ and $\mathcal{M} = \{m\}$, we can use (123) to show that the condition

$$\max_{m \in [n]} \left\| \hat{\mathbf{R}}_X^m - \mathbf{R}_X^m \right\|_2 < \eta_* \, , \tag{125}$$

ensures that

$$\mathsf{dim\,col}\left( \mathbf{R}_X^m \right) = \mathsf{dim\,col}\left( \hat{\mathbf{R}}_X^m; \eta \right) \, . \tag{126}$$

(ii) For any generic $\mathcal{M}$, we can use (124) to show that the condition

$$\max_{m \in [n]} \left\| \hat{\mathbf{R}}_X^m - \mathbf{R}_X^m \right\|_2 < \frac{\eta_*}{n} \, , \tag{127}$$

ensures that

$$\mathsf{dim\,col}\left( \mathbf{R}_X^{\mathcal{M}} \right) = \mathsf{dim\,col}\left( \hat{\mathbf{R}}_X^{\mathcal{M}}; \eta \right) \, . \tag{128}$$

This concludes the proof. ■

Since we have a sufficient condition to ensure that approximate ranks of $\{\hat{\mathbf{R}}_X^{\mathcal{M}} : \mathcal{M} \subseteq [n]\}$ are equal to the ranks of $\{\mathbf{R}_X^{\mathcal{M}} : \mathcal{M} \subseteq [n]\}$, now we can state a sufficient condition for finding a valid causal order.

**Lemma 14** (Bounded noise – Causal order). *Let $\eta \in (0, \eta^*)$. If*

$$\max_{m \in [n]} \left\| \hat{\mathbf{R}}_X^m - \mathbf{R}_X^m \right\|_2 < \frac{\eta_*}{n} \, , \tag{129}$$

*then $I \circ \pi$ is a valid causal order, where $\pi$ is the output of Algorithm 1.*

*Proof:* We note that Algorithm 1 only uses approximate rank statements for $\{\hat{\mathbf{R}}_X^{\mathcal{M}} : \mathcal{M} \subseteq [n]\}$. Therefore, if the approximate rank statements are equal to the ranks of $\{\mathbf{R}_X^{\mathcal{M}} : \mathcal{M} \subseteq [n]\}$, then, the infinite sample guarantees of Algorithm 1 provided in Lemma 9 immediately transfer to the noisy $\mathbf{R}_X^m$ regime. ■

Next, we note that Algorithms 2 and 3 use the *direction* information of the approximate column and null spaces of $\{\hat{\mathbf{R}}_X^{\mathcal{M}} : \mathcal{M} \subseteq [n]\}$ as well. Therefore, we need to bound the "distance" of the estimated subspaces from their noise-free counterparts. The main results we will use for this purpose are the Davis–Kahan $\sin\theta$ theorems [19], specialized for approximate column and null spaces.

**Lemma 15** ([19], $\sin\theta$ theorem). *Consider two $k \times k$ real symmetric matrices $\mathbf{A}$ and $\hat{\mathbf{A}}$. Denote the orthonormal bases of the column and null spaces of $\mathbf{A}$ by $\mathbf{A}_1$ and $\mathbf{A}_0$, respectively, such that*

$$\mathbf{A} = \mathbf{A}_1 \mathbf{C}_1 \mathbf{A}_1^\top \, , \tag{130}$$

*where $\mathbf{C}_1$ is a symmetric matrix with the same eigenvalues as $\mathbf{A}$. Similarly, denote the orthonormal bases for the approximate column and null space of $\hat{\mathbf{A}}$ by $\hat{\mathbf{A}}_1$ and $\hat{\mathbf{A}}_0$, such that*

$$\hat{\mathbf{A}} = \hat{\mathbf{A}}_0 \hat{\mathbf{C}}_0 \hat{\mathbf{A}}_0^\top + \hat{\mathbf{A}}_1 \hat{\mathbf{C}}_1 \hat{\mathbf{A}}_1^\top \, , \tag{131}$$

*where eigenvalues of $\hat{\mathbf{C}}_0$ are that of the approximate null space of $\hat{\mathbf{A}}$, and the eigenvalues of $\hat{\mathbf{C}}_1$ are that of the approximate column space of $\hat{\mathbf{A}}$. Define*

$$\chi \triangleq \min \left\{ \boldsymbol{\lambda}_{\min}(\hat{\mathbf{C}}_1), \, \boldsymbol{\lambda}_{\min}(\mathbf{C}_1) - \boldsymbol{\lambda}_{\max}(\hat{\mathbf{C}}_0) \right\}, \tag{132}$$

*where $\boldsymbol{\lambda}_{\max}$ and $\boldsymbol{\lambda}_{\min}$ denote the maximum and minimum eigenvalue of a symmetric matrix. Then, given that the approximate rank of $\hat{\mathbf{A}}$ is equal to the rank of $\mathbf{A}$ and $\chi > 0$, we have*

$$\left\| \hat{\mathbf{A}}_0^\top \cdot \mathbf{A}_1 \right\|_2 \leq \frac{1}{\chi} \|\hat{\mathbf{A}} - \mathbf{A}\|_2, \tag{133}$$

*and*

$$\left\| \hat{\mathbf{A}}_1^\top \cdot \mathbf{A}_0 \right\|_2 \leq \frac{1}{\chi} \|\hat{\mathbf{A}} - \mathbf{A}\|_2. \tag{134}$$

**Lemma 16** ([19, Symmetric $\sin\theta$ theorem]). *Under the setting described in the $\sin\theta$ theorem above, we have*

$$\left\| \hat{\mathbf{A}}_0 \cdot \hat{\mathbf{A}}_0^\top - \mathbf{A}_0 \cdot \mathbf{A}_0^\top \right\|_2 \leq \frac{1}{\chi} \|\hat{\mathbf{A}} - \mathbf{A}\|_2, \tag{135}$$

*and*

$$\left\| \hat{\mathbf{A}}_1 \cdot \hat{\mathbf{A}}_1^\top - \mathbf{A}_1 \cdot \mathbf{A}_1^\top \right\|_2 \leq \frac{1}{\chi} \|\hat{\mathbf{A}} - \mathbf{A}\|_2. \tag{136}$$

We note that $\chi$ in these $\sin\theta$ theorems can be bounded using Weyl's inequality (Lemma 12) and the minimum nonzero eigenvalue of matrix $\mathbf{A}$ as follows.

$$\boldsymbol{\lambda}_{\max}(\hat{\mathbf{C}}_0) \leq \|\hat{\mathbf{A}} - \mathbf{A}\|_2, \qquad \boldsymbol{\lambda}_{\min}(\hat{\mathbf{C}}_1) \geq \boldsymbol{\lambda}_{\min}(\mathbf{C}_1) - \|\hat{\mathbf{A}} - \mathbf{A}\|_2. \tag{137}$$

Therefore,

$$\chi \geq \boldsymbol{\lambda}_{\min}(\mathbf{C}_1) - \|\hat{\mathbf{A}} - \mathbf{A}\|_2. \tag{138}$$

Using $\mathbf{R}_X^{\mathcal{M}}$ for $\mathbf{A}$ and $\hat{\mathbf{R}}_X^{\mathcal{M}}$ for $\hat{\mathbf{A}}$, and using the definition of $\eta^*$ in (110), we obtain

$$\chi \geq \eta^* - \left\| \hat{\mathbf{R}}_X^{\mathcal{M}} - \mathbf{R}_X^{\mathcal{M}} \right\|_2. \tag{139}$$

In order to ensure that the approximate rank of $\hat{\mathbf{R}}_X^{\mathcal{M}}$ is equal to the rank of $\mathbf{R}_X^{\mathcal{M}}$, it suffices that $\|\hat{\mathbf{R}}_X^{\mathcal{M}} - \mathbf{R}_X^{\mathcal{M}}\|_2$ satisfies

$$\eta_* > \left\| \hat{\mathbf{R}}_X^{\mathcal{M}} - \mathbf{R}_X^{\mathcal{M}} \right\|_2, \tag{140}$$

due to (121). If this holds, we get

$$\chi > \eta^* - \eta_* = \eta^* - \min\{\eta, \eta^* - \eta\} = \max\{\eta, \eta^* - \eta\} \geq \min\{\eta, \eta^* - \eta\} = \eta_*. \tag{141}$$

Therefore, the left-hand sides of (133)–(136) are upper bounded by

$$\frac{1}{\eta_*} \left\| \hat{\mathbf{R}}_X^{\mathcal{M}} - \mathbf{R}_X^{\mathcal{M}} \right\|_2. \tag{142}$$

We can use Davis–Kahan symmetric $\sin\theta$ theorem (Lemma 16) to prove that encoder estimation error is upper bounded by the error in $\hat{\mathbf{R}}_X^m$.

**Lemma 17** (Bounded noise – Encoder). *Let $\eta \in (0, \eta^*)$. If (125) holds, then the output $\mathbf{H}$ of Algorithm 3 satisfies*

$$\mathbf{H} \cdot \mathbf{G} = \mathbf{P}_I \cdot (\mathbf{C}_{\mathrm{pa}} + \mathbf{C}_{\mathrm{err}}), \tag{143}$$

*where for all $i \notin \overline{\mathrm{pa}}(j)$, $\mathbf{C}_{\mathrm{pa}}$ satisfies $(\mathbf{C}_{\mathrm{pa}})_{i,j} = 0$, and*

$$\left\| \mathbf{C}_{\mathrm{err}} \right\|_2 \leq \frac{\sqrt{n}}{\eta_*} \cdot \left\| \mathbf{G} \right\|_2 \cdot \max_{m \in [n]} \left\| \hat{\mathbf{R}}_X^m - \mathbf{R}_X^m \right\|_2. \tag{144}$$

*Proof:* We prove this result by considering each row $m$ of $\mathbf{H}$ separately. In Algorithm 3, we select $\mathbf{H}_{m,:}$ from $\mathrm{col}(\hat{\mathbf{R}}_X^m; \eta)$ for each $m \in [n]$. We start by noting that in Lemma 7, we have shown that using the noise-free score difference correlation matrices, we get

$$\left( \mathbf{v}^\top \cdot \mathbf{G} \right)_i \neq 0 \implies i \in \overline{\mathrm{pa}}(I^m) \qquad \forall \mathbf{v} \in \mathrm{col}(\mathbf{R}_X^m), \, \|\mathbf{v}\|_2 = 1. \tag{145}$$

We can decompose any $\mathbf{v} \in \mathrm{col}(\hat{\mathbf{R}}_X^m; \eta)$ as

$$\mathbf{v} = \mathbf{A}_0 \cdot \mathbf{A}_0^\top \cdot \mathbf{v} + \mathbf{A}_1 \cdot \mathbf{A}_1^\top \cdot \mathbf{v}, \tag{146}$$

where $\mathbf{A}_0$ and $\mathbf{A}_1$ are the orthonormal bases of the null and column spaces of $\mathbf{R}_X^m$ in line with the notation of the Davis–Kahan $\sin \theta$ theorem. Note that $\mathbf{A}_1 \cdot \mathbf{A}_1^\top$ is the orthogonal projection operator onto the column space of $\mathbf{R}_X^m$, therefore, using (145), we have

$$\left( (\mathbf{A}_1 \cdot \mathbf{A}_1^\top \cdot \mathbf{v})^\top \cdot \mathbf{G} \right)_i \neq 0 \implies i \in \overline{\mathrm{pa}}(I^m) \qquad \forall \mathbf{v} \in \mathrm{col}(\hat{\mathbf{R}}_X^m; \eta), \ \|\mathbf{v}\|_2 = 1. \tag{147}$$

Therefore, the "erroneous" entries in $\mathbf{v}^\top \cdot \mathbf{G}$ can be written as

$$\left( \mathbf{v}^\top \cdot \mathbf{G} \right)_i = \left( (\mathbf{A}_0 \cdot \mathbf{A}_0^\top \cdot \mathbf{v})^\top \cdot \mathbf{G} \right)_i \qquad \forall i \notin \overline{\mathrm{pa}}(I^m), \ \forall \mathbf{v} \in \mathrm{col}(\hat{\mathbf{R}}_X^m; \eta), \ \|\mathbf{v}\|_2 = 1. \tag{148}$$

Note that if we choose $\mathbf{v} \in \mathrm{col}(\hat{\mathbf{R}}_X^m; \eta)$ as row $m$ of $\mathbf{H}$, we have

$$\left\| (\mathbf{P}_I \cdot \mathbf{C}_{\mathrm{err}})_{m,:} \right\|_2^2 = \sum_{i \notin \overline{\mathrm{pa}}(I^m)} \left( \mathbf{v}^\top \cdot \mathbf{G} \right)_i^2 \leq \left\| (\mathbf{A}_0 \cdot \mathbf{A}_0^\top \cdot \mathbf{v})^\top \cdot \mathbf{G} \right\|_2^2. \tag{149}$$

Since $\mathbf{v} \in \mathrm{col}(\hat{\mathbf{R}}_X^m; \eta)$, we have $\hat{\mathbf{A}}_1 \cdot \hat{\mathbf{A}}_1^\top \cdot \mathbf{v} = \mathbf{v}$, where $\hat{\mathbf{A}}_1$ is the orthonormal basis of the approximate column space of $\hat{\mathbf{R}}_X^m$. Using this and $\|\mathbf{v}\|_2 = 1$, we can write (149) as

$$\left\| (\mathbf{P}_I \cdot \mathbf{C}_{\mathrm{err}})_{m,:} \right\|_2 \leq \left\| \mathbf{A}_0 \cdot \mathbf{A}_0^\top \cdot \mathbf{v} \right\|_2 \cdot \left\| \mathbf{G} \right\|_2 \tag{150}$$

$$= \left\| \mathbf{A}_0 \cdot \mathbf{A}_0^\top \cdot \hat{\mathbf{A}}_1 \cdot \hat{\mathbf{A}}_1^\top \cdot \mathbf{v} \right\|_2 \cdot \left\| \mathbf{G} \right\|_2 \tag{151}$$

$$\leq \left\| \mathbf{A}_0 \cdot \mathbf{A}_0^\top \cdot \hat{\mathbf{A}}_1 \cdot \hat{\mathbf{A}}_1^\top \right\|_2 \cdot \left\| \mathbf{G} \right\|_2 \tag{152}$$

$$= \left\| \mathbf{A}_0^\top \cdot \hat{\mathbf{A}}_1 \right\|_2 \cdot \left\| \mathbf{G} \right\|_2, \tag{153}$$

Using Davis–Kahan $\sin \theta$ theorem (Lemma 15), when (125) holds, we can further upper bound the $m$-th row of the error term by

$$\left\| (\mathbf{P}_I \cdot \mathbf{C}_{\mathrm{err}})_{m,:} \right\|_2 \leq \frac{1}{\eta_*} \left\| \hat{\mathbf{R}}_X^m - \mathbf{R}_X^m \right\|_2 \cdot \left\| \mathbf{G} \right\|_2. \tag{154}$$

Note that the spectral norm is upper bounded by the Frobenius norm. Therefore,

$$\left\| \mathbf{C}_{\mathrm{err}} \right\|_2 \leq \left\| \mathbf{C}_{\mathrm{err}} \right\|_{\mathrm{F}} \leq \sqrt{n} \cdot \frac{1}{\eta_*} \cdot \left\| \mathbf{G} \right\|_2 \cdot \max_{m \in [n]} \left\| \hat{\mathbf{R}}_X^m - \mathbf{R}_X^m \right\|_2. \tag{155}$$

∎

Next, we investigate the approximate orthogonality test used for Algorithm 2 under noisy score difference correlation matrices.

**Lemma 18.** *Let* $\mathbf{A}, \mathbf{B}, \hat{\mathbf{A}}, \hat{\mathbf{B}}$ *be* $k \times k$ *symmetric matrices, such that*

$$\mathbf{A} = \mathbf{A}_0 \mathbf{C}_0 \mathbf{A}_0^\top + \mathbf{A}_1 \mathbf{C}_1 \mathbf{A}_1^\top, \qquad \hat{\mathbf{A}} = \hat{\mathbf{A}}_0 \hat{\mathbf{C}}_0 \hat{\mathbf{A}}_0^\top + \hat{\mathbf{A}}_1 \hat{\mathbf{C}}_1 \hat{\mathbf{A}}_1^\top, \tag{156}$$

$$\mathbf{B} = \mathbf{B}_0 \mathbf{D}_0 \mathbf{B}_0^\top + \mathbf{B}_1 \mathbf{D}_1 \mathbf{B}_1^\top, \qquad \hat{\mathbf{B}} = \hat{\mathbf{B}}_0 \hat{\mathbf{D}}_0 \hat{\mathbf{B}}_0^\top + \hat{\mathbf{B}}_1 \hat{\mathbf{D}}_1 \hat{\mathbf{B}}_1^\top, \tag{157}$$

*where* $[\mathbf{A}_0, \mathbf{A}_1]$, $[\mathbf{B}_0, \mathbf{B}_1]$, $[\hat{\mathbf{A}}_0, \hat{\mathbf{A}}_1]$, $[\hat{\mathbf{B}}_0, \hat{\mathbf{B}}_1]$ *are all* $k \times k$ *orthogonal matrices, and shapes of* $\mathbf{A}_0$–$\hat{\mathbf{A}}_0$ *and* $\mathbf{B}_0$–$\hat{\mathbf{B}}_0$ *match. Given that the spectra of (i)* $\mathbf{C}_0$ *and* $\hat{\mathbf{C}}_1$*, (ii)* $\mathbf{C}_1$ *and* $\hat{\mathbf{C}}_0$*, (iii)* $\mathbf{D}_0$ *and* $\hat{\mathbf{D}}_1$*, and (iv)* $\mathbf{D}_1$ *and* $\hat{\mathbf{D}}_0$ *are all separated with gap* $\chi$*, we have*

*(i) If* $\|\mathbf{A}_0^\top \cdot \mathbf{B}_0\|_2 = 0$*, that is, if* $\mathrm{col}(\mathbf{A}_0) \perp \mathrm{col}(\mathbf{B}_0)$*, then,*

$$\left\| \hat{\mathbf{A}}_0^\top \cdot \hat{\mathbf{B}}_0 \right\|_2 \leq \frac{1}{\chi} \left( \left\| \hat{\mathbf{A}} - \mathbf{A} \right\|_2 + \left\| \hat{\mathbf{B}} - \mathbf{B} \right\|_2 \right). \tag{158}$$

*(ii) If* $\|\mathbf{A}_0^\top \cdot \mathbf{B}_0\|_2 = \mu \neq 0$*, that is,* $\mathrm{col}(\mathbf{A}_0) \not\perp \mathrm{col}(\mathbf{B}_0)$*, then,*

$$\left\| \hat{\mathbf{A}}_0^\top \cdot \hat{\mathbf{B}}_0 \right\|_2 \geq \mu - \frac{1}{\chi} \left( \left\| \hat{\mathbf{A}} - \mathbf{A} \right\|_2 + \left\| \hat{\mathbf{B}} - \mathbf{B} \right\|_2 \right). \tag{159}$$

*Proof:* We first prove a perturbation bound on the product of orthogonal projectors.

$$\left\| \hat{\mathbf{A}}_0 \cdot \hat{\mathbf{A}}_0^\top \cdot \hat{\mathbf{B}}_0 \cdot \hat{\mathbf{B}}_0^\top - \mathbf{A}_0 \cdot \mathbf{A}_0^\top \cdot \mathbf{B}_0 \cdot \mathbf{B}_0^\top \right\|_2 \tag{160}$$

$$= \left\| \hat{\mathbf{A}}_0 \cdot \hat{\mathbf{A}}_0^\top \cdot \left( \hat{\mathbf{B}}_0 \cdot \hat{\mathbf{B}}_0^\top - \mathbf{B}_0 \cdot \mathbf{B}_0^\top \right) - \left( \mathbf{A}_0 \cdot \mathbf{A}_0^\top - \hat{\mathbf{A}}_0 \cdot \hat{\mathbf{A}}_0^\top \right) \cdot \mathbf{B}_0 \cdot \mathbf{B}_0^\top \right\|_2 \tag{161}$$

$$\leq \left\| \hat{\mathbf{A}}_0 \cdot \hat{\mathbf{A}}_0^\top \right\|_2 \cdot \left\| \hat{\mathbf{B}}_0 \cdot \hat{\mathbf{B}}_0^\top - \mathbf{B}_0 \cdot \mathbf{B}_0^\top \right\|_2 + \left\| \mathbf{A}_0 \cdot \mathbf{A}_0^\top - \hat{\mathbf{A}}_0 \cdot \hat{\mathbf{A}}_0^\top \right\|_2 \cdot \left\| \mathbf{B}_0 \cdot \mathbf{B}_0^\top \right\|_2 \tag{162}$$

$$\leq \left\| \hat{\mathbf{B}}_0 \cdot \hat{\mathbf{B}}_0^\top - \mathbf{B}_0 \cdot \mathbf{B}_0^\top \right\|_2 + \left\| \mathbf{A}_0 \cdot \mathbf{A}_0^\top - \hat{\mathbf{A}}_0 \cdot \hat{\mathbf{A}}_0^\top \right\|_2 . \tag{163}$$

Using Davis–Kahan symmetric $\sin\theta$ theorem (Lemma 16) yields

$$\left\| \hat{\mathbf{A}}_0 \cdot \hat{\mathbf{A}}_0^\top \cdot \hat{\mathbf{B}}_0 \cdot \hat{\mathbf{B}}_0^\top - \mathbf{A}_0 \cdot \mathbf{A}_0^\top \cdot \mathbf{B}_0 \cdot \mathbf{B}_0^\top \right\|_2 \leq \frac{1}{\chi} \left( \left\| \hat{\mathbf{A}} - \mathbf{A} \right\|_2 + \left\| \hat{\mathbf{B}} - \mathbf{B} \right\|_2 \right) . \tag{164}$$

We use this result in two separate cases: $\mathsf{col}(\mathbf{A}_0) \perp \mathsf{col}(\mathbf{B}_0)$ and $\mathsf{col}(\mathbf{A}_0) \not\perp \mathsf{col}(\mathbf{B}_0)$.

 (i) **Case 1:** $\mathsf{col}(\mathbf{A}_0) \perp \mathsf{col}(\mathbf{B}_0)$. In this case, (164) immediately yields (158).

 (ii) **Case 2:** $\mathsf{col}(\mathbf{A}_0) \not\perp \mathsf{col}(\mathbf{B}_0)$. Let us define $\|\mathbf{A}_0^\top \cdot \mathbf{B}_0\|_2 = \mu \neq 0$. We can use (164) and triangle inequality to get

$$\left\| \hat{\mathbf{A}}_0 \cdot \hat{\mathbf{A}}_0^\top \cdot \hat{\mathbf{B}}_0 \cdot \hat{\mathbf{B}}_0^\top \right\|_2 + \frac{1}{\chi} \left( \left\| \hat{\mathbf{A}} - \mathbf{A} \right\|_2 + \left\| \hat{\mathbf{B}} - \mathbf{B} \right\|_2 \right) \tag{165}$$

$$\geq \left\| \hat{\mathbf{A}}_0 \cdot \hat{\mathbf{A}}_0^\top \cdot \hat{\mathbf{B}}_0 \cdot \hat{\mathbf{B}}_0^\top \right\|_2 + \left\| \hat{\mathbf{A}}_0 \cdot \hat{\mathbf{A}}_0^\top \cdot \hat{\mathbf{B}}_0 \cdot \hat{\mathbf{B}}_0^\top - \mathbf{A}_0 \cdot \mathbf{A}_0^\top \cdot \mathbf{B}_0 \cdot \mathbf{B}_0^\top \right\|_2 \tag{166}$$

$$\geq \left\| \mathbf{A}_0 \cdot \mathbf{A}_0^\top \cdot \mathbf{B}_0 \cdot \mathbf{B}_0^\top \right\|_2 \tag{167}$$

$$= \left\| \mathbf{A}_0^\top \cdot \mathbf{B}_0 \right\|_2 = \mu . \tag{168}$$

Rearranging the terms yields (159).

∎

Finally, we show that the graph estimation is correct under bounded noise.

**Lemma 19** (Bounded noise – Graph). *Let* $\eta \in (0, \eta^*)$ *and* $\gamma \in (0, \gamma^*)$, *where*

$$\gamma^* \triangleq \min_{t \in [n]} \left\| \mathbf{G}_{:,t} \right\|_2 \cdot \left\| (\mathbf{G}^\dagger)_{t,:} \right\|_2 , \quad and \quad \gamma_* \triangleq \min\{\gamma, \gamma^* - \gamma\} . \tag{169}$$

*Under Assumption 1, the output* $\hat{\mathcal{G}}$ *of Algorithm 2 is the graph isomorphism of the transitive closure of the true latent graph* $\mathcal{G}$ *if*

$$\max_{m \in [n]} \left\| \hat{\mathbf{R}}_X^m - \mathbf{R}_X^m \right\|_2 < \min \left\{ \frac{\eta_*}{n}, \frac{\eta_* \gamma_*}{2n} \right\} . \tag{170}$$

*Proof:* Algorithm 2 uses approximate column and null spaces of $\mathbf{R}_X^{\mathcal{M}}$ for different $\mathcal{M} \subseteq [n]$. Therefore, the generic sufficient condition (113), that is,

$$\max_{m \in [n]} \left\| \hat{\mathbf{R}}_X^m - \mathbf{R}_X^m \right\|_2 < \frac{\eta_*}{n} , \tag{171}$$

ensures that the approximate ranks of all $\hat{\mathbf{R}}_X^{\mathcal{M}}$ matrices investigated in the algorithm will be equal to the rank of $\mathbf{R}_X^{\mathcal{M}}$. To ensure that the estimated graph will be correct, we must guarantee that the approximate orthogonality tests will also give correct results. Using Lemma 18, we can show that the test metric is upper bounded in terms of $\max_{m \in [n]} \|\hat{\mathbf{R}}_X^m - \mathbf{R}_X^m\|_2$. We do so by setting $\mathbf{A} = \mathbf{R}_X^{\mathcal{V}_t}$, $\hat{\mathbf{A}} = \hat{\mathbf{R}}_X^{\mathcal{V}_t}$, $\mathbf{B} = \mathbf{R}_X^{\mathcal{M}_{t,j}}$, $\hat{\mathbf{B}} = \hat{\mathbf{R}}_X^{\mathcal{M}_{t,j}}$; and $\mathbf{A}_0$ as the orthonormal basis of the column space of $\mathbf{A}$, and $\mathbf{B}_0$ as the null space of $\mathbf{B}$. As shown in (141), (113) ensures that the spectral gap $\chi$ in Lemma 18 statement is at least $\chi > \eta_*$. In Lemma 10, we show that in the infinite sample, noise-free regime, when investigating $(t, j)$, the test metric is given by

$$\left\| \mathbf{A}_0^\top \cdot \mathbf{B}_0 \right\|_2 = \begin{cases} 0 & I^t \to I^j \in \mathcal{G} , \\ \|\mathbf{G}_{:,t}\|_2 \cdot \|(\mathbf{G}^\dagger)_{t,:}\|_2 & I^t \not\to I^j \in \mathcal{G} . \end{cases} \tag{172}$$

Since the approximate orthogonality test needs to distinguish even the weakest instance of non-orthogonality from orthogonality, we define

$$\gamma^* \triangleq \min_{t \in [n]} \left\| \mathbf{G}_{:,t} \right\|_2 \cdot \left\| (\mathbf{G}^\dagger)_{t,:} \right\|_2 . \tag{173}$$

Using Lemma 18, we see that for $(t, j)$ for which $I^t \to I^j$ exists in $\mathcal{G}$, if (113) holds, then

$$\left\| \hat{\mathbf{A}}_0^\top \cdot \hat{\mathbf{B}}_0 \right\|_2 < \frac{2n}{\eta_*} \max_{m \in [n]} \left\| \hat{\mathbf{R}}_X^m - \mathbf{R}_X^m \right\|_2 . \tag{174}$$

Similarly, for $(t, j)$ for which $I^t \to I^j$ does not exist in $\mathcal{G}$, if (113) holds, then

$$\left\| \hat{\mathbf{A}}_0^\top \cdot \hat{\mathbf{B}}_0 \right\|_2 > \gamma^* - \frac{2n}{\eta_*} \max_{m \in [n]} \left\| \hat{\mathbf{R}}_X^m - \mathbf{R}_X^m \right\|_2 . \tag{175}$$

Therefore, for a choice of $\gamma \in (0, \gamma^*)$ to yield correct orthogonality decisions, it suffices that

$$\frac{2n}{\eta_*} \max_{m \in [n]} \left\| \hat{\mathbf{R}}_X^m - \mathbf{R}_X^m \right\|_2 < \gamma < \gamma^* - \frac{2n}{\eta_*} \max_{m \in [n]} \left\| \hat{\mathbf{R}}_X^m - \mathbf{R}_X^m \right\|_2 , \tag{176}$$

or, equivalently,

$$\frac{2n}{\eta_*} \max_{m \in [n]} \left\| \hat{\mathbf{R}}_X^m - \mathbf{R}_X^m \right\|_2 < \min\{\gamma, \gamma^* - \gamma\} \triangleq \gamma_* . \tag{177}$$

In other words, when using $\gamma \in (0, \gamma^*)$ as the approximate orthogonality threshold, we can correctly identify orthogonal subspaces from non-orthogonal ones when

$$\max_{m \in [n]} \left\| \hat{\mathbf{R}}_X^m - \mathbf{R}_X^m \right\|_2 < \frac{\eta_* \gamma_*}{2n} . \tag{178}$$

Due to Lemma 10, if all the approximate rank and orthogonality tests are correct, then the output $\hat{\mathcal{G}}$ of Algorithm 2 is the graph isomorphism of the transitive closure of the true latent graph $\mathcal{G}$. ∎

## D   Sample complexity for generic consistent score estimator

In this section, we will transform the bounded noise identifiability guarantees in Appendix C to sample complexity statements about a generic consistent score difference estimator. Specifically, recall the consistent score difference estimator model from (15).

$$\mathbb{P}\left( \max_{m \in [n]} \left\| \hat{\boldsymbol{d}}_X^m(\cdot; \mathcal{X}_N) - \boldsymbol{d}_X^m \right\|_{p_X} \le \epsilon \right) \ge 1 - \delta , \qquad \forall N \ge N(\epsilon, \delta) . \tag{179}$$

Recall the definitions of the constants in (24),

$$\beta \triangleq \left( 4 \max_{m \in [n]} \left\| \boldsymbol{d}_X^m \right\|_{p_X} \right)^{-1} \qquad \text{and} \qquad \beta_{\min} \triangleq 2 \min_{m \in [n]} \left\| \boldsymbol{d}_X^m \right\|_{p_X} . \tag{180}$$

First, we derive the sufficient samples for $(\epsilon, \delta)$-approximating $\hat{\mathbf{R}}_X^m$ using Lemma 11.

**Lemma 20.** *For any $\epsilon > 0$ and $\delta > 0$, using $N_R(\epsilon, \delta)$ samples suffice to ensure that*

$$\mathbb{P}\left( \max_{m \in [n]} \left\| \hat{\mathbf{R}}_X^m - \mathbf{R}_X^m \right\|_2 \le \epsilon \right) \ge 1 - \delta , \qquad \forall N \ge N_R(\epsilon, \delta) , \tag{181}$$

*where*

$$N_R(\epsilon, \delta) \triangleq N\left( \min\{\epsilon \cdot \beta, \beta_{\min}\}, \delta \right) . \tag{182}$$

*Proof:* In Lemma 11, in order to bound the error in $\hat{\mathbf{R}}_X^m$ for a specific $m \in [n]$, we require

$$\left\| \boldsymbol{d}_X^m - \hat{\boldsymbol{d}}_X^m \right\|_{p_X} < 2 \cdot \left\| \boldsymbol{d}_X^m \right\|_{p_X} . \tag{183}$$

To ensure this for all $m \in [n]$, it suffices that

$$\max_{m \in [n]} \left\| \hat{\boldsymbol{d}}_X^m - \boldsymbol{d}_X^m \right\|_{p_X} < 2 \min_{m \in [n]} \left\| \boldsymbol{d}_X^m \right\|_{p_X} \triangleq \beta_{\min} . \tag{184}$$

Under this condition, Lemma 11 shows that the error in correlation matrices $\mathbf{R}_X^m$ is bounded by the error in $\boldsymbol{d}_X^m$ as

$$\left\| \hat{\mathbf{R}}_X^m - \mathbf{R}_X^m \right\|_2 \le 4 \cdot \left\| \boldsymbol{d}_X^m \right\|_{p_X} \cdot \left\| \hat{\boldsymbol{d}}_X^m - \boldsymbol{d}_X^m \right\|_{p_X} . \tag{185}$$

To get an $m$-agnostic bound, we can take $\max$ of both sides to get

$$\max_{m \in [n]} \left\| \hat{\mathbf{R}}_X^m - \mathbf{R}_X^m \right\|_2 \leq 4 \cdot \max_{m \in [n]} \left\| \boldsymbol{d}_X^m \right\|_{p_X} \cdot \max_{m \in [n]} \left\| \hat{\boldsymbol{d}}_X^m - \boldsymbol{d}_X^m \right\|_{p_X} . \tag{186}$$

In other words, to achieve $\epsilon$ error in estimating $\hat{\mathbf{R}}_X^m$, the error in estimating $\boldsymbol{d}_X^m$ suffices to be upper bounded by

$$\epsilon \cdot \beta \triangleq \frac{\epsilon}{4 \max_{m \in [n]} \left\| \boldsymbol{d}_X^m \right\|_{p_X}} > \max_{m \in [n]} \left\| \hat{\boldsymbol{d}}_X^m - \boldsymbol{d}_X^m \right\|_{p_X} . \tag{187}$$

In summary, (184) and (187) together gives that

$$\max_{m \in [n]} \left\| \hat{\boldsymbol{d}}_X^m - \boldsymbol{d}_X^m \right\|_{p_X} < \min \left\{ \beta_{\min}, \epsilon \cdot \beta \right\} , \tag{188}$$

suffices to ensure

$$\max_{m \in [n]} \left\| \hat{\mathbf{R}}_X^m - \mathbf{R}_X^m \right\|_2 \leq \epsilon . \tag{189}$$

Using the sample complexity of $\hat{\boldsymbol{d}}_X^m$, this means that

$$N \geq N_R(\epsilon, \delta) \triangleq N \left( \min \left\{ \epsilon \cdot \beta, \beta_{\min} \right\}, \delta \right) \tag{190}$$

samples suffice to ensure this error upper bound on $\hat{\mathbf{R}}_X^m$ with probability at least $1 - \delta$. ∎

Next, we restate and prove the sample complexity of estimating the causal order in Lemma 4.

**Lemma 4** (Sample complexity – Causal order). *Let $\eta \in (0, \eta^*)$. Under Assumption 1, for any $\delta > 0$, $N_{\mathrm{rank}}(\delta)$ samples suffice to ensure that with probability at least $1 - \delta$, $I \circ \pi$ is a valid causal order, where $\pi$ is the output of Algorithm 1, where*

$$N_{\mathrm{rank}}(\delta) \triangleq N \left( \min \left\{ \frac{\beta \eta_*}{n}, \beta_{\min} \right\}, \delta \right) . \tag{191}$$

*Proof:* Lemma 14 states that as long as $\hat{\mathbf{R}}_X^m$ error is upper bounded by $\frac{\eta_*}{n}$, the output of Algorithm 1 is correct. Then, using Lemma 20, we see that

$$N \geq N_R \left( \frac{\eta_*}{n}, \delta \right) = N \left( \min \left\{ \frac{\beta \eta_*}{n}, \beta_{\min} \right\}, \delta \right) \tag{192}$$

samples suffice to ensure that the estimated causal order is correct with probability at least $1 - \delta$. ∎

Next, we prove the sample complexity of estimating the latent graph.

**Theorem 1** (Sample complexity – Graph). *Let $\eta \in (0, \eta^*)$ and $\gamma \in (0, \gamma^*)$. Under Assumption 1, for any $\delta > 0$, $N_{\mathcal{G}}(\delta)$ samples suffice to ensure that collective output $\hat{\mathcal{G}}(\mathcal{X}_N)$ of Algorithms 1 and 2 satisfies $\delta$–PAC graph recovery, where*

$$N_{\mathcal{G}}(\delta) \triangleq N \left( \min \left\{ \frac{\beta \eta_*}{n}, \frac{\beta \eta_* \gamma_*}{2n}, \beta_{\min} \right\}, \delta \right) . \tag{193}$$

*Proof:* Lemma 19 states that the estimated graph is correct if

$$\max_{m \in [n]} \left\| \hat{\mathbf{R}}_X^m - \mathbf{R}_X^m \right\|_2 < \min \left\{ \frac{\eta_*}{n}, \frac{\eta_* \gamma_*}{2n} \right\} . \tag{194}$$

Therefore, Lemma 20 implies that

$$N \geq N \left( \min \left\{ \frac{\beta \eta_*}{n}, \frac{\beta \eta_* \gamma_*}{2n}, \beta_{\min} \right\}, \delta \right) \tag{195}$$

samples suffice for correct graph recovery with probability at least $1 - \delta$. ∎

Similarly, we prove the sample complexity of estimating the causal variables.

**Theorem 2** (Sample complexity – Variables). *Let $\eta \in (0, \eta^*)$. For any $\epsilon > 0$ and $\delta > 0$, $N_Z(\epsilon, \delta)$ samples suffice to ensure that the output $\mathbf{H}(\mathcal{X}_N)$ of Algorithm 3 satisfies $(\epsilon, \delta)$–PAC causal variables recovery, where*

$$N_Z(\epsilon, \delta) \triangleq N \left( \min \left\{ \frac{\epsilon \beta \eta_*}{\sqrt{n} \|\mathbf{G}\|_2}, \beta \eta_*, \beta_{\min} \right\}, \delta \right) . \tag{196}$$

*Proof:* Lemma 17 states that if the input error is bounded by

$$\max_{m \in [n]} \left\| \hat{\mathbf{R}}_X^m - \mathbf{R}_X^m \right\|_2 < \eta_* \, , \tag{197}$$

then the estimated encoder achieves error rate $\epsilon$

$$\left\| \mathbf{C}_{\text{err}} \right\|_2 \leq \sqrt{n} \cdot \frac{1}{\eta_*} \cdot \left\| \mathbf{G} \right\|_2 \cdot \max_{m \in [n]} \left\| \hat{\mathbf{R}}_X^m - \mathbf{R}_X^m \right\|_2 \leq \epsilon \, . \tag{198}$$

Therefore, to ensure $(\epsilon, \delta)$–PAC causal variables recovery, it suffices that, with probability at least $1 - \delta$, the input error on $\hat{\mathbf{R}}_X^m$ is upper bounded by

$$\max_{m \in [n]} \left\| \hat{\mathbf{R}}_X^m - \mathbf{R}_X^m \right\|_2 < \min \left\{ \frac{\epsilon \eta_*}{\sqrt{n} \| \mathbf{G} \|_2}, \eta_* \right\} \, . \tag{199}$$

Using Lemma 20, this implies that

$$N \geq N \left( \min \left\{ \frac{\epsilon \beta \eta_*}{\sqrt{n} \| \mathbf{G} \|_2}, \beta \eta_*, \beta_{\min} \right\}, \delta \right) \tag{200}$$

samples suffice for $(\epsilon, \delta)$–PAC causal variables recovery. ∎

# E   Sample complexity for RKHS-based score estimator

In this section, we adopt the RKHS-based score estimator of [12], state its assumptions, derive its noise model for score *difference* estimation, and use this model to make the sample complexity upper bounds derived in Appendix D explicit.

The score estimator in [12] requires the support of the distribution to be an open subset of its domain [12, Assumption B.1]. Therefore, we estimate the score function of $X$ through its orthogonal projection to the $n$ dimensional support manifold $\text{col}(\mathbf{G})$. The projection matrix can be constructed from observed data samples with probability 1 as long as $N \geq n$.

Next, we state Lemma 5 formally including all the assumptions required.

**Lemma 5** (Formal). *Denote a trace-class matrix-valued kernel by $\mathcal{K} \colon \mathbb{R}^d \times \mathbb{R}^d \to \mathbb{R}^{d \times d}$, denote its associated RKHS by $\mathcal{H}_\mathcal{K}$, and define $\kappa \triangleq \sup_{x \in \text{col}(\mathbf{G})} \text{tr} \mathcal{K}(x, x)$. Define an integral operator on $\mathcal{H}_\mathcal{K}$ as*

$$L_{\mathcal{K}, p} f \triangleq \int \mathcal{K}(x, \cdot) f(x) p(x) \, \mathrm{d}x \, , \quad \forall f \in \mathcal{H}_\mathcal{K} \, , \tag{201}$$

*where $p$ is a pdf. Assume that there exists $f^0 \in \mathcal{H}_\mathcal{K}$ and $f^m \in \mathcal{H}_\mathcal{K}$ for all $m \in [n]$ such that $\boldsymbol{s}_X = L_{\mathcal{K}, p_X} f^0$ and $\boldsymbol{s}_X^m = L_{\mathcal{K}, p_X^m} f^m$. Then, using Tikhonov regularization, under [12, Assumptions B.1–B.5], for any $m \in [n]$, for any $\delta \in (0, 1)$ and $N \geq (2\sqrt{2}\kappa^2 \log 8n/\delta)^4$, with probability at least $1 - \delta$, score estimator in [12, Theorem B.1] satisfies*

$$\max_{m \in [n]} \left\| \hat{\boldsymbol{d}}_X^m - \boldsymbol{d}_X^m \right\|_{p_X} \leq C \cdot N^{-1/4} \cdot \log 8n/\delta \, , \tag{202}$$

*where $C$ is a sample independent constant that depends only on $p_X$, $p_X^m$ for $m \in [n]$ and the structure of $\mathcal{H}_\mathcal{K}$.*

*Proof:* We use the fact that for a function $f \in \mathcal{H}_\mathcal{K}$, we have

$$\left\| f \right\|_{p_X} = \left\| L_{\mathcal{K}, p_X} f \right\|_{\mathcal{H}_\mathcal{K}} \leq \left\| L_{\mathcal{K}, p_X} \right\|_{\text{op}} \cdot \left\| f \right\|_{\mathcal{H}_\mathcal{K}} \, . \tag{203}$$

Therefore, even if we estimate $\boldsymbol{s}_X^m$ using data from $p_X^m$ but evaluate it on data from $p_X$, we can still bound the MSE of the estimate using the Hilbert norm. The result states that the Hilbert norm error bound is, for any $m \in \{0\} \cup [n]$, and if $N \geq (2\sqrt{2}\kappa^2 \log 4/\delta)^4$,

$$\mathbb{P} \left( \left\| \hat{\boldsymbol{s}}_X^m - \boldsymbol{s}_X^m \right\|_{\mathcal{H}_\mathcal{K}} \leq C_1 \cdot N^{-1/4} \cdot \log \frac{4}{\delta} \right) \geq 1 - \delta \, . \tag{204}$$

Since $\| \cdot \|_{p_X}$ is a norm, we have

$$\left\| \hat{\boldsymbol{d}}_X^m - \boldsymbol{d}_X^m \right\|_{p_X} \leq \left\| \hat{\boldsymbol{s}}_X^m - \boldsymbol{s}_X^m \right\|_{p_X} + \left\| \hat{\boldsymbol{s}}_X - \boldsymbol{s}_X \right\|_{p_X} \leq \left\| L_{\mathcal{K}, p_X} \right\|_{\text{op}} \cdot \left( \left\| \hat{\boldsymbol{s}}_X^m - \boldsymbol{s}_X^m \right\|_{\mathcal{H}_\mathcal{K}} + \left\| \hat{\boldsymbol{s}}_X - \boldsymbol{s}_X \right\|_{\mathcal{H}_\mathcal{K}} \right) \, . \tag{205}$$

Combining with the Hilbert norm error bound, taking union bound and defining $C \triangleq 2C_1 \|L_{\mathcal{K},p_X}\|_{\mathrm{op}}$, we get

$$\mathbb{P}\left( \|\hat{\boldsymbol{d}}_X^m - \boldsymbol{d}_X^m\|_{p_X} \leq C \cdot N^{-1/4} \cdot \log\frac{4}{\delta} \right) \geq 1 - 2\delta \,. \tag{206}$$

Note that the noise model in (15) requires a bound on $\max_{m \in [n]} \|\hat{\boldsymbol{d}}_X^m - \boldsymbol{d}_X^m\|_{p_X}$. By taking a union bound, we get

$$\mathbb{P}\left( \max_{m \in [n]} \|\hat{\boldsymbol{d}}_X^m - \boldsymbol{d}_X^m\|_{p_X} \leq C \cdot N^{-1/4} \cdot \log\frac{4}{\delta} \right) \geq 1 - 2n\delta \,. \tag{207}$$

By replacing $\delta$ with $\delta/2n$, we get, when $N \geq (2\sqrt{2}\kappa^2 \log 8n/\delta)^4$,

$$\mathbb{P}\left( \max_{m \in [n]} \|\hat{\boldsymbol{d}}_X^m - \boldsymbol{d}_X^m\|_{p_X} \leq C \cdot N^{-1/4} \cdot \log\frac{8n}{\delta} \right) \geq 1 - \delta \,, \tag{208}$$

which concludes the proof. ∎

**Corollary 2.** *The error bound given in this result can be immediately transformed into a sample complexity statement. Specifically,*

$$\mathbb{P}\left( \max_{m \in [n]} \|\hat{\boldsymbol{d}}_X^m - \boldsymbol{d}_X^m\|_{p_X} \leq \epsilon \right) \geq 1 - \delta \,, \quad \forall N \geq N(\epsilon, \delta) \,, \tag{209}$$

*where*

$$N(\epsilon, \delta) = \left( \max\left\{ 2\sqrt{2}\kappa^2, \frac{C}{\epsilon} \right\} \right)^4 \cdot \left( \log\frac{8n}{\delta} \right)^4 \,. \tag{210}$$

Using this $N(\cdot, \cdot)$ function, we can immediately make the sample complexity results in the previous section explicit. The theorem statements are also stated here for completeness.

**Theorem 3** (RKHS-based sample complexity – Graph). *Let $\eta \in (0, \eta^*)$ and $\gamma \in (0, \gamma^*)$. Under Assumption 1 and the conditions of Lemma 5, $N_{\mathcal{G}}(\delta)$ samples sufficient to ensure that the collective output $\hat{\mathcal{G}}(\mathcal{X}_N)$ of Algorithms 1 and 2 satisfies $\delta$–PAC graph recovery, where*

$$N_{\mathcal{G}}(\delta) = \left( \max\left\{ \frac{C}{\epsilon_{\mathcal{G}}}, 2\sqrt{2}\kappa^2 \right\} \right)^4 \cdot \left( \log\frac{8n}{\delta} \right)^4 \,, \quad \text{and} \quad \epsilon_{\mathcal{G}} \triangleq \min\left\{ \frac{\beta\eta_*}{n}, \frac{\beta\eta_*\gamma_*}{2n}, \beta_{\min} \right\} \,. \tag{211}$$

**Theorem 4** (RKHS-based sample complexity – Variables). *Let $\eta \in (0, \eta^*)$. Under the conditions of Lemma 5, $N_Z(\epsilon, \delta)$ samples ensure that the output $\mathbf{H}(\mathcal{X}_N)$ of Algorithm 3 satisfies $(\epsilon, \delta)$–PAC causal variables recovery, where*

$$N_Z(\epsilon, \delta) = \left( \max\left\{ \frac{C}{\epsilon \cdot \epsilon_Z}, 2\sqrt{2}\kappa^2 \right\} \right)^4 \cdot \left( \log\left( \frac{8n}{\delta} \right) \right)^4 \,, \tag{212}$$

*and*

$$\epsilon_Z \triangleq \min\left\{ \frac{\epsilon\beta\eta_*}{\sqrt{n}\|\mathbf{G}\|_2}, \beta\eta_*, \beta_{\min} \right\} \,. \tag{213}$$

## F  Properties of positive semidefinite matrices

In this section, we present properties of positive semidefinite (PSD) matrices that are commonly used in our proofs.

**Proposition 2.** *For a real $k \times k$ PSD matrix $\mathbf{A}$, the null space of $\mathbf{A}$ can be specified using the quadratic form as*

$$\mathsf{null}(\mathbf{A}) = \left\{ \mathbf{v} \in \mathbb{R}^k \,:\, \mathbf{v}^\top \mathbf{A}\mathbf{v} = 0 \right\} \,. \tag{214}$$

*Proof:* For a vector $\mathbf{v} \in \mathbb{R}^k$, $\mathbf{A}\mathbf{v} = 0$ implies $\mathbf{v}^\top \mathbf{A}\mathbf{v} = 0$ unconditionally. We prove the reverse direction by showing that $\mathbf{A}\mathbf{v} \neq 0$ implies $\mathbf{v}^\top \mathbf{A}\mathbf{v} \neq 0$. First, since $\mathbf{A}$ is PSD, it has a Cholesky decomposition $\mathbf{A} = \mathbf{U}^\top \mathbf{U}$ for some matrix $\mathbf{U} \in \mathbb{R}^{k \times k}$. If $\mathbf{A}\mathbf{v} = \mathbf{U}^\top \mathbf{U}\mathbf{v} \neq 0$, then $\mathbf{U}\mathbf{v} \neq 0$. This implies $\mathbf{v}^\top \mathbf{A}\mathbf{v} = \|\mathbf{U}\mathbf{v}\|_2^2 \neq 0$. This concludes the proof. ∎

**Proposition 3.** *If* $\mathbf{A}, \mathbf{B}$ *are real* $k \times k$ *PSD matrices, then*

$$\mathsf{null}(\mathbf{A} + \mathbf{B}) = \mathsf{null}(\mathbf{A}) \cap \mathsf{null}(\mathbf{B}) \,, \tag{215}$$

*and*

$$\mathsf{col}(\mathbf{A} + \mathbf{B}) = \mathsf{col}(\mathbf{A}) + \mathsf{col}(\mathbf{B}) \,, \tag{216}$$

*where* $+$ *denotes the* Minkowski sum *between sets, which is defined, for any two subsets* $\mathcal{A}$ *and* $\mathcal{B}$ *of a vector space, as*

$$\mathcal{A} + \mathcal{B} \triangleq \left\{ \mathbf{a} + \mathbf{b} \,:\, \mathbf{a} \in \mathcal{A}, \, \mathbf{b} \in \mathcal{B} \right\} . \tag{217}$$

*Proof:* Using Proposition 2, we can write $\mathsf{null}(\mathbf{A} + \mathbf{B})$ as

$$\mathsf{null}(\mathbf{A} + \mathbf{B}) = \left\{ \mathbf{v} \in \mathbb{R}^k \,:\, \mathbf{v}^\top (\mathbf{A} + \mathbf{B}) \mathbf{v} = 0 \right\} . \tag{218}$$

Since $\mathbf{A}$ and $\mathbf{B}$ are both PSD, for any $\mathbf{v} \in \mathbb{R}^k$, we have $\mathbf{v}^\top \mathbf{A} \mathbf{v} \geq 0$ and $\mathbf{v}^\top \mathbf{B} \mathbf{v} \geq 0$. Therefore, $\mathbf{v}^\top (\mathbf{A} + \mathbf{B}) \mathbf{v} = 0$ holds if and only if $\mathbf{v}^\top \mathbf{A} \mathbf{v} = 0$ and $\mathbf{v}^\top \mathbf{B} \mathbf{v} = 0$. That is,

$$\mathsf{null}(\mathbf{A} + \mathbf{B}) = \left\{ \mathbf{v} \in \mathbb{R}^k \,:\, \mathbf{v}^\top \mathbf{A} \mathbf{v} = 0 \,\wedge\, \mathbf{v}^\top \mathbf{B} \mathbf{v} = 0 \right\} . \tag{219}$$

We note that, due to Proposition 2, the right hand side of this expression is equal to $\mathsf{null}(\mathbf{A}) \cap \mathsf{null}(\mathbf{B})$, which concludes the proof for (215).

Next, note that the column space of a symmetric matrix is the orthogonal complement of its null space. Then, using (215), we have

$$\mathsf{col}(\mathbf{A} + \mathbf{B}) = (\mathsf{null}(\mathbf{A} + \mathbf{B}))^\perp = \left(\mathsf{null}(\mathbf{A}) \cap \mathsf{null}(\mathbf{B})\right)^\perp . \tag{220}$$

Since $\mathsf{null}(\mathbf{A})$ and $\mathsf{null}(\mathbf{B})$ are subspaces of a finite dimensional vector space, we have

$$(\mathsf{null}(\mathbf{A}) \cap \mathsf{null}(\mathbf{B}))^\perp = \mathsf{null}(\mathbf{A})^\perp + \mathsf{null}(\mathbf{B})^\perp = \mathsf{col}(\mathbf{A}) + \mathsf{col}(\mathbf{B}) \,. \tag{221}$$

This concludes the proof for (216). ∎

# G  Hyperparameter estimation

In this section, we provide minor modifications to our algorithms that enable constructing rough estimates for threshold upper bounds $\eta^*$ and $\gamma^*$ in practice.

**Estimating** $\eta^*$ : We note that in Algorithm 1, at step $t$, all matrices investigated must have either rank $t$ or $t - 1$. Therefore, the $t - 1$-th eigenvalues of all of these matrices are non-zero, or equivalently we have $\geq \eta^*$. This observation lets us to iteratively build and refine an upper bound on $\eta^*$ during Algorithm 1, which we can use as a surrogate of $\eta^*$ in the subsequent Algorithm 2 and 3. For Algorithm 1 itself, at time $t$, we can pick $k \in \mathcal{V}_t$ with minimal $t$-th eigenvalue—this is a threshold-free way of estimating the causal order.

**Estimating** $\gamma^*$ : We first note that definition in (23) depends entirely on the ground truth decoder matrix $\mathbf{G}$ and its pseudoinverse. Secondly, we note that Algorithm 3 can be used independently of Algorithm 2 to generate an estimate $\mathbf{H}$ of $\mathbf{G}^\dagger$ up to some estimation error and possible mixing among rows. We can compute the value

$$\min_{i \in [n]} \|\mathbf{H}_{i,:}\|_2 \cdot \|\mathbf{H}^\dagger_{:,i}\|_2 \,, \tag{222}$$

which would be equal to $\gamma^*$ had $\mathbf{H} = \mathbf{G}^\dagger$, and use it as our estimate of $\gamma^*$.

