# OpenReview forum: "Sample Complexity of Interventional Causal Representation Learning"
_NeurIPS.cc/2024/Conference — NeurIPS 2024 poster_

### Official Review · Reviewer_fPsb · 2024-07-11

**Soundness:** 3
**Presentation:** 3
**Contribution:** 3
**Rating:** 7
**Confidence:** 2

**Summary:**

Interventional causal representation learning aims to recover the causal graph for latent variables and simultaneously recover the latent variables. While identifiability has been established for the infinite sample regime, this is not practical. This paper aims to provide PAC-style bounds on identifiability (of both the graph and latent variables) in the finite-sample regime, assuming stochastic soft interventions and linear transformations from the latent to the observation space. Three prototypical algorithms (causal order estimation, graph estimation, and inverse transformation estimation) are proposed to estimate the graph and the latent variables, which pave the foundation, from which  PAC-style bounds have been established. Numerical assessments are conducted to provide complementary insights into the sample complexity results.

**Strengths:**

Identifiability analysis is often conducted in the infinite sample regime, limiting its practical applicability. This paper extends identifiability analysis to the finite sample regime, representing a significant step forward.

**Weaknesses:**

The bounds are rather loose, which limits its practical applicability. Apart from that, these bounds do not provide sufficient guidance on designing better algorithms recovering latent variables and graphs. Having said so, I still welcome this type of theoretical papers.

**Questions:**

I wonder if causal order estimation is necessary. Since the latent variables have yet to be estimated, their values and semantics are flexible. This implies that any given causal order could be acceptable, as these variables can be learned to fit the specified orders. Some edges would be pruned later, which can be done in the graph estimation stage.

Additionally, I wonder if there are other quantities beyond score differences and the correlation of score differences that could be used to recover the graph and the latent variables.

Furthermore, it would be beneficial if the analysis could assess which variables and which parts of the graph are more prone to error, and to what degree, providing feedback on the estimated/recovered variables and graphs.

**Limitations:**

n.a.

---

> ### Author Rebuttal · Authors · 2024-08-06
>
> We thank the reviewer for the kind review and thoughtful questions. We address the raised questions as follows.
>
> **Causal order estimation.** The reviewer is correct in that since the causal graph and variables are latent, we can choose any ordering of the nodes and latent variables first and then assign meanings later. In fact, in Line 98, without loss of generality, we say that $(1,\dots,n)$ is a causal order. However, after fixing this assignment, *we do not know* which node intervened in which environment (e.g., we do not know the causal order between the nodes intervened in $\mathcal{E}^i$ and $\mathcal{E}^j$). Hence, in our algorithms, we specifically identify each interventional environment with the corresponding intervened latent variable. For example, in Algorithm 3, we use the $m$-th interventional environment where node $k$ is intervened to recover the intervened variable $Z_k$ (up to some mixing) and assign it specifically to the $m$-th component of our latent variables estimate $\hat{Z}$.
>
>  **Approaches to CRL and score-based framework.**
> - There exist different approaches to interventional CRL that don’t use score differences. One such example is [9], in which the authors present a sparsity-promoting VAE-based approach.
>
> - Our motivation for investigating the score difference-based approach stems from the fact that it provides the strongest existing results for multiple settings. For the most general setting of nonparametric transforms and causal models, the only known provably correct algorithm is score-based [7]. For linear transformations, [4] provides the most general results (without restricting the causal models).
>
> - On an interesting note, the approach of [6] for Gaussian latent models and linear transformations is based on precision matrix differences, which exactly correspond to score function differences between Gaussian distributions.
>
> **Point-error analysis.** In our algorithms, we use thresholded decisions to obtain our graph estimate. Therefore, as a detection problem, the decisions that are most likely to fail are those with values that, in the noise-free setting, are closest to the decision boundary, i.e., the thresholds. In our algorithms, it is possible to keep track of the estimated values of these statistics as surrogates of the noise-free values and therefore assess how error-prone each decision is. That being said, the analysis in our paper is sequential, and errors can propagate. We consider the sample complexity analysis of approximate latent graph recovery, up to $k$-point errors, a critical future direction.

---

> > ### Comment · Reviewer_fPsb · 2024-08-09
> >
> > I'm satisfied with the response and keep the score as 'accept.'

---

### Official Review · Reviewer_Q5cw · 2024-07-12

**Soundness:** 3
**Presentation:** 3
**Contribution:** 3
**Rating:** 7
**Confidence:** 3

**Summary:**

This paper establishes finite sample guarantees for recovering the latent causal graph under stochastic soft interventions and observations under a unknown linear transformation. Experiments are conducted and code is given.

**Strengths:**

The paper is well motivated and the contributions are clear.

**Weaknesses:**

I did not check the proofs in detail but I do not spot any glaring weaknesses.

The experiments are rather small scale (up to 10 nodes).

**Questions:**

- What is $P_I$ in equation (12)?
- In practice, do we need to know $\eta^*$ or $\gamma^*$ to invoke your results? How do we know enough samples have been collected?

**Limitations:**

Nil

---

> ### Author Rebuttal · Authors · 2024-08-06
>
> We thank the reviewer for the kind review and thoughtful questions.
>
> - ${\bf P}_I$ is the permutation matrix for permutation $I$, the intervention order. We will include the definition in the notations paragraph.
>
> -  The current theorem statements are given for specific choice of thresholds that require the knowledge of unknown model parameters. However, this is **not necessary** for our analysis – it suffices to choose $\eta\in(0,\eta^*)$ and $\gamma\in(0,\gamma^*)$, which results in similar sample complexity bounds that depend on the thresholds $\eta$ and $\gamma$ directly. Since this flexibility is critical for the practicality of our results, we will modify our theorem statements to make them clear. We emphasize that the changes to the proofs and the results are minuscule: it suffices to replace all occurrences of $\eta^*/2$ and $\gamma^*/2$ in our sample complexity statements with $\min\\\{\eta,\eta^*-\eta\\\}$ and $\min\\\{\gamma,\gamma^*-\gamma\\\}$, respectively. For details, please see our response to reviewer jaoy on “Setting thresholds”.
>
> - For implementation purposes, this means that we can select $\eta$ and $\gamma$ arbitrarily small. We highlight that requiring hyperparameters (i.e., $\eta$ and $\gamma$) to be restricted in an interval determined by unknown model parameters (i.e., $\eta^*$ and $\gamma^*$ in our algorithm) is standard (and generally inevitable) in the analysis of regularized algorithms in high dimensional statistics for even simpler problems like sparse support recovery. A common example is the error bound for regularized lasso problem in [HTW, Theorem 11.1], which requires the regularization parameter $\lambda_N$ to be greater than $2\\|{\bf X}^\top{\bf w}\\|_{\infty}/N$ – which depends on the _noise realizations_ ${\bf w}$.
>
> [HTW] T.Hastie, R.Tibshirani, and M.Wainwright. Statistical learning with sparsity. _Monographs on statistics and applied probability_, 143, 2015.
>
> - *Number of latent nodes*: We kindly note that the existing interventional CRL literature generally works with small graphs: $n=10$ is the largest graph size considered in the closely related single-node interventional CRL literature (e.g., for linear transformations [6] considers 5 nodes, [4] considers 8 nodes, and [11] considers 10 nodes. For linear causal models under nonlinear mixing Buchholz et al. (2023) consider 5 nodes. For fully nonparametric CRL, [13] considers 2 nodes, and [7] considers 8 nodes). Therefore, our experiments with up to 10 nodes are on par with the state-of-the-art interventional CRL.
>
> S. Buchholz, G. Rajendran, E. Rosenfeld, B. Aragam, B. Schölkopf, and P. Ravikumar. Learning
> linear causal representations from interventions under general nonlinear mixing. NeurIPS 2023.

---

> > ### Comment · Reviewer_Q5cw · 2024-08-09
> >
> > Thank you for your responses!
> >
> > I understand from your response that one can select $\eta$ and $\gamma$ to be arbitrarily small for implementation. However, from my understanding of your theoretical claims, this should blow up the required number of samples in order to conclude "something useful", right? Can you give concrete suggestions on how practitioners can use your algorithm while drawing meaningful conclusions?

---

> > > ### Author Response · Authors · 2024-08-09
> > >
> > > Thanks for the thoughtful question. In practice, we can include routines for estimating $\eta^*$ and $\gamma^*$. These estimates do not have to be highly accurate and even rough estimates suffice to choose reliable thresholds $\eta$ and $\gamma$.  Specifically, based on estimates for $\eta^*$ and $\gamma^*$  we can choose  “safe” thresholds, e.g., one-fourth of the estimates, so that (i) with high probability we satisfy the requirement $\eta < \eta^*$, and (ii) we can avoid collecting excessive samples and compromising the sample complexity bounds. For practical purposes, we can construct such estimates for both $\eta^*$ and $\gamma^*$ as follows.
> > >
> > > - **Estimate for $\eta^{*}$:** We note that in Algorithm 1, at step $t$, all matrices investigated must have  either rank $t$ or $t-1$. Therefore, the $t-1$-th eigenvalues of all of these matrices are non-zero, or equivalently we have  $\geq\eta^*$. This observation lets us to iteratively build and refine an upper bound on $\eta^*$ during Algorithm 1, which we can use as a surrogate of $\eta^*$ in the subsequent algorithms 2 and 3. For Algorithm 1 itself, at time $t$, we can pick $k\in{\cal V}_t$ with minimal $t$-th eigenvalue—this is a threshold-free way of estimating the causal order, but might not be amenable to tractable analysis. In order to keep the analysis more amenable to sample complexity analysis, we can use the current statement of Algorithm 1 with thresholded rank statements.
> > >
> > > - **Estimate for $\gamma^{*}$:** We first note that $\gamma^*$ definition in eq.(23) depends entirely on the ground truth decoder matrix $\bf G$ and its pseudoinverse. Secondly, we note that Algorithm 3 can be used independently of Algorithm 2 to generate an estimate $\bf H$ of ${\bf G}^\dagger$ up to some estimation error and possible mixing among rows. We can compute the value $\min\_{i\in[n]}\\|{\bf H}\_{i,:}\\|_2\cdot\\|{\bf H}^\dagger\_{:,i}\\|\_2$, which would be equal to $\gamma^*$ had ${\bf H} = {\bf G}^\dagger$, and use it as our estimate of $\gamma^*$.

---

> > > > ### Comment · Reviewer_Q5cw · 2024-08-13
> > > >
> > > > Thank you for the response. I will maintain my positive score.

---

### Official Review · Reviewer_jaoy · 2024-07-13

**Soundness:** 2
**Presentation:** 3
**Contribution:** 3
**Rating:** 6
**Confidence:** 3

**Summary:**

This paper provides finite-sample identifiability results for recovering the latent causal graph and the generating latent variables given observations in a high-dimensional space that have been generated by the latent variables by a linear transformation and single-node soft interventional data with interventions on all nodes. Previous work provides identifiability results only in the infinite-sample regime. The algorithms rely on score-based estimators which have been studied in literature. Specifically the correlation matrix of the score differences are shown to be constrained by the graph and the linear transformation matrix. To recover the graph, first a topological order on the variables is estimated. Verifying whether a particular edge exists is checked by a necessary and sufficient condition on the approximate orthogonality of the column space and null space. For the variables recovery, it is sufficient to consider a unit vector in the column space. An existing RKHS-based score estimator is used to obtain the final sample-complexity results.

**Strengths:**

Causal representation learning is an active field of study and theoretical work that provides finite-sample identifiability results is quite relevant in practice and of interest to the community. Overall the paper is well-written with a clear flow of ideas that makes it easy to follow.

I liked that the proposed method has a decoupling between the score estimators and the rest of the algorithms with proofs for the infinite-sample regime separate. This helps us to understand that the difference in quality of finite-sample data versus infinite sample data is the score-estimators.

**Weaknesses:**

A few weaknesses (with the major one being point 1) follow:

1. A major confusion for me was that the setting of the thresholds was not clear. If the true scores are not known, then how is \eta set from \eta* which depends on the unknown true score estimates. Similarly for \gamma that depends on \gamma^* that depends on the true transformation matrix. What am I missing?

2. Overall, the paper has multiple instances of notations not defined which make the paper difficult to parse. For example, Definition 2 has P_I, C_{pa}'s diagonal entries undefined? Lemma 2 has the pseudoinverse subscript undefined.

3. I also think that the experimental section needs basic details. One suggestion would be to cut out the final RKHS sample complexity bounds that are just plugging (29) into Theorem 1 and 2.

4. The proofs in the paper first derive results for the infinite-sample regime where score estimates are error-free. Finite-sample guarantees are then obtained by bounding the error in the score estimates. It would make the paper stronger if the infinite-sample regime results are compared with existing work.

5. In Figure 1b, why are there spikes in the recovery rate even for worse score estimate MSE? Does this go against the main message of the paper that worse score estimates should imply poorer graph recovery and latent variable estimation?

6. What is the reason behind defining the variables recovery as it is defined in Definition 2. Is there a previous instance of such a definition because this doesn't seem standard. Any rationale would help the reader.

Overall, I think the paper is a solid contribution if these questions are clarified. But my current score can't improve given these questions.

**Questions:**

Please refer to the weakness section for major issues. A few miscellanous points:
1) I haven't checked the details but it struck me as weird to notice that there is a specific value of \gamma in (20) that verifies whether an edge exists or not. Is there any intuition for this?
2) Typos - (22) M \subseteq [n], Line 179.
3) In Figure 1b, is there a reason why even for better score estimates, there is no point for the 10 nodes case?

**Limitations:**

Implicitly addressed in the conclusions.

---

> ### Author Rebuttal · Authors · 2024-08-06
>
> We thank the reviewer for a thorough review and thoughtful questions. We hope our answers address the main concerns of the reviewer.
>
> **Setting the thresholds.** We are grateful for the reviewer bringing this up. We recognize that our choices of thresholds could have been presented differently and in a significantly more general way. As the reviewer correctly points out, the current theorem statements are given for specific choice of thresholds that require the knowledge of unknown model parameters. However, this is **not necessary** for our analysis – it suffices to choose $\eta\in (0,\eta^*)$ and $\gamma\in(0,\gamma^*)$, which results in similar sample complexity bounds that depend on the thresholds $\eta$ and $\gamma$ directly. We will modify our theorem statements to make them clear. Let us explain why this choice suffices.
>
> - We prove our sample complexity results by deriving sufficient upper bounds on the score estimation noise for our thresholded approximate tests to succeed. We use two tests: a) An approximate rank test with soft threshold $\eta$ that uses Weyl’s inequality for the analysis and b) An approximate subspace orthogonality test with soft threshold $\gamma$ that uses Davis Kahan (symmetric) sin $\Theta$ theorem for analysis.
>
> - In both cases, the approximate test uses a metric $\chi$ to distinguish between two models, one where the noise-free metric value is $\chi = 0$ vs the one where it is at least $\chi\geq\eta^*$ (or $\gamma^*$). In order to guarantee the correct output using a threshold $\eta$ (or $\gamma$), we must ensure that maximal error $\Delta$ for the metric $\chi$ satisfies $\Delta\leq\min\\{\eta,\eta^*-\eta\\}$ (similarly for $\gamma$). Sample complexity results directly follow from these upper bounds on $\Delta$.
>
> - For implementation purposes, this means that we can select $\eta$ and $\gamma$ arbitrarily small. We highlight that requiring hyperparameters (i.e., $\eta$ and $\gamma$) to be restricted in an interval determined by unknown model parameters (i.e., $\eta^*$ and $\gamma^*$ in our algorithm) is standard (and generally inevitable) in the analysis of regularized algorithms in high dimensional statistics for even simpler problems like sparse support recovery. A common example is the error bound for regularized lasso problem in [HTW, Theorem 11.1], which requires the regularization parameter $\lambda_N$ to be greater than $2\\|{\bf X}^\top{\bf w}\\|_{\infty}/N$ – which depends on the _noise realizations_ $\bf w$.
>
> [HTW] T.Hastie, R.Tibshirani, and M.Wainwright. Statistical learning with sparsity. _Monographs on statistics and applied probability_, 143, 2015.
>
> **Experiment details.** The experimental details are currently provided in Appendix G. We will add those details to the main paper by using the additional page that we can have in the final version.
>
> We also note that Theorems 3 and 4 are essential in the main body. It’s correct that the main results are Theorems 1 and 2, yet they are implicit in the sample complexity of a score estimator. We believe that having Theorems 3 and 4 explicitly is essential for accessibility.
>
> **Experiment results.** Each data point in Figure 1b corresponds to a specific $(N,n,d)$ tuple (listed in Appendix G). Importantly, for each $(N, n)$, the current plot includes two $d$ values: $n$ and 15. In x axis of Figure 1b, we erroneously plotted the MSE divided by $d$, which resulted in a plot that is the union of two shifted monotonic graphs for each $n$. We have corrected this error in Figure 1 of the PDF attached to the global response and presented the results in two figures accordingly. The new plot shows the desired monotonic behavior without major performance spikes.
>
> In the same new plot, we also added a data point for a larger number of samples to show that a good graph recovery rate is possible for latent dimension $n=10$.
>
> **Infinite sample results.** Thanks for the suggestion. We will add the following theorem (and its latent variables recovery counterpart), along with the attendant discussion, to the main paper.
>
> _**Theorem (Infinite samples – Graph).**
> In the infinite sample regime, under Assumption 1, for any $\eta\in(0,\eta^*)$ and $\gamma\in(0,\gamma^*)$, the collective output $\hat{\mathcal{G}}$ of Algorithms 1 and 2 is the graph isomorphism of the transitive closure of the true latent graph $\mathcal{G}$._
>
> **Latent variable recovery definition.**  In the interventional CRL literature under linear transforms, two types of results exist. First, the so-called “perfect recovery” in which the latent variables are recovered up to permutations and scaling, which is defined exactly in the manner of our Definition 2 with ${\bf C}\_{\rm err} = 0$ and ${\bf C}\_{\rm pa}$ being a diagonal matrix, e.g., Definition 4 in [4]; Theorem 5.3 in [5]; Definition 1 in [9]. Secondly, the recovered latent variables are linear functions of some other variables as well. This recovery class is natural since Theorem 6 of [11] shows that element-wise latent variable identifiability is _impossible_ under soft interventions. Then, the various identifiability classes are defined similarly to Definition 2, e.g., Definition 5 in [4]. In these definitions, off-diagonal support of the recovery matrices (e.g.,  ${\bf C}\_{\rm pa}$) denotes the mixing with other random variables that vary under different results, e.g., we consider “mixing up to parents” whereas [3] and [4] consider up to “mixing with ancestors”.
>
> **Notations and typos.** We thank the reviewer for pointing out several issues regarding notations and typos. ${\bf P}_I$ is the permutation matrix corresponding to $I$, the intervention order. We will carefully fix the other under-explained notations and correct the typos in the revised paper.

---

> > ### Comment · Reviewer_jaoy · 2024-08-13
> > **Response to rebuttal**
> >
> > The response to the reviewer below addresses my concern about the setting of thresholds. It seems like there still is a need to estimate the true \eta^* and \gamma^* practically. I am satisfied with the other responses and have increased my score accordingly.

---

### Official Review · Reviewer_Yb2M · 2024-07-13

**Soundness:** 3
**Presentation:** 3
**Contribution:** 3
**Rating:** 7
**Confidence:** 3

**Summary:**

This paper provides finite sample analysis for causal representation learning with general latent causal models, single-node soft interventions, and linear mixing functions. Sample complexity for identifying graphs up to ancestors and identifying latent variables up to mixing with parent variables have been studied.

**Strengths:**

1. To the best of my knowledge, this is the first finite sample analysis for interventional CRL, which is a very interesting problem with high practical value.

2. All theoretical results including definitions are organized clearly.

**Weaknesses:**

1. The considered CRL setting is a little bit restrictive compared to the recent ones. Specifically, the assumption of linear mixing function may restrict its usage in a lot of real-world scenarios. Meanwhile, the indeterminacy is non-trivial, and further process/disentanglement might still be needed. However, it is understandable since sample complexity might be much more difficult to get for more general models.

2. Perhaps it would be better if there was more discussion on the assumptions.

**Questions:**

1. How to make sure that all the imperfections in the experiments are due to score estimates?

2. Are there any intuitive ideas to extend the range of the finite sample analysis to more general settings in CRL?

**Limitations:**

Yes.

---

> ### Author Rebuttal · Authors · 2024-08-06
>
> We thank the reviewer for finding our results interesting and the thoughtful questions. We address the questions as follows.
>
> - **Source of error in experiments.**
> To assess the source of errors in experiments, we have the following analysis: In Appendix B, we establish the identifiability guarantees under the infinite-sample (i.e., perfect scores) regime. To verify this, we perform experiments using ground truth score functions. In these experiments, the graph estimation results were perfectly accurate in 100 runs, which implies that the graph errors we report in the paper are due to score estimation. We will add these perfect score results to the paper. To replicate these experiments with ground truth score functions, one can set the `estimate_score_fn` flag in `finite_sample_test.py` file line 57 to `False` and run the file.
>
> - **Intuitions for finite sample analysis for general CRL.** In this paper, we have shown that some critical decision problems relevant to CRL – such as edge detection – can be done
> via soft decisions in a way that is amenable to error analysis. While the specific tools used (e.g., subspace estimation) were limited to CRL under linear transformations, we believe that the overarching approach of reducing CRL to soft decisions and then obtaining finite sample guarantees of each such step is a good starting point for designing and analyzing more general CRL algorithms.

---

> > ### Comment · Reviewer_Yb2M · 2024-08-14
> >
> > Thank you for the response. I will maintain my positive rating.

---

### Author Rebuttal · Authors · 2024-08-06

In the attached PDF, we provide a figure to update Figure 1b of the submitted paper with the updated experiment results. Specifically, there are two changes:
1) Each data point in old Figure 1b corresponds to a specific tuple $(N,n,d)$ (listed in Appendix G). For each $(N,n)$, the old plot has two points of $d$: $d=n$ and $d=15$. In the new figure, we present the two cases of $d$ in two figures for a clear presentation. The new plots show a monotonic behavior without major spikes.
2) In the new plots, we also added a data point for a larger number of samples to show that a good graph recovery rate is possible for latent dimension $n=10$.

---

### Decision · Program_Chairs · 2024-09-25

**Decision:**

Accept (poster)

**Comment:**

The paper provides the first finite sample complexity results for causal representation learning, albeit under some technical conditions.

All reviewers were supportive of accepting the paper.

For a camera-ready version, please take into the account the comments from all reviewers regarding clarifications, for instance: notation/typos, discussion on the assumptions, experiment details and discussion about causal order estimation.